# Eco-Friendly Silver Nanoparticles Synthesis Method Using Medicinal Plant Fungal Endophytes—Biological Activities and Molecular Docking Analyses

**DOI:** 10.3390/biology14080950

**Published:** 2025-07-28

**Authors:** Harish Chandra, Sagar Vishwakarma, Nilesh Makwana, Arun S. Kharat, Vijeta Chaudhry, Sumit Chand, Rajendra Prasad, Soban Prakash, Annapurna Katara, Archana Yadav, Manisha Nigam, Abhay Prakash Mishra

**Affiliations:** 1Department of Botany and Microbiology, Gurukula Kangri (Deemed to be University), Haridwar 249404, India; sagarvishwakarmarksh@gmail.com (S.V.); chaudhryvijeta@gmail.com (V.C.); sct19930415@gmail.com (S.C.); annapurnakatara@gmail.com (A.K.); 2Laboratory of Applied Microbiology and Cancer Remedies, School of Life Sciences, Jawaharlal Nehru University, New Delhi 110067, India; mnr2808@gmail.com (N.M.); arunkharat2007@gmail.com (A.S.K.); 3School of Agriculture, Uttaranchal University, Dehradun 248007, India; rajenpd@gmail.com; 4High Altitude Plant Physiology Research Centre (HAPPRC), Hemvati Nandan Bahuguna Garhwal University, Srinagar Garhwal 246174, India; sobanprakash87@gmail.com; 5Department of Microbiology, Institute of Biosciences and Biotechnology, Chhatrapati Shahu Ji Maharaj University, Kanpur 208024, India; archana25578@gmail.com; 6Department of Biochemistry, School of Life Sciences, Hemvati Nandan Bahuguna Garhwal (Central) University, Srinagar Garhwal 246174, India; m.nigam@hnbgu.ac.in; 7Cosmetics and Natural Product Research Center (CosNat), Department of Pharmaceutical Sciences, Naresuan University, Phitsanulok 65000, Thailand

**Keywords:** green synthesis, silver nanoparticles, endophytic fungi, antibacterial activity, anticancer activity, ADME, toxicity, sustainable nanotechnology

## Abstract

This work uses *Corynespora smithii*, an endophytic fungus found in *Bergenia ciliata* leaves, to investigate the environmentally friendly manufacture of silver nanoparticles (AgNPs). The use of dangerous chemicals was avoided, since the fungal extracts served as stabilizing and reducing agents. Strong biological activity, such as antibacterial, anticancer, and antioxidant properties, were demonstrated by the produced AgNPs. Both Gram-positive and Gram-negative bacteria were efficiently inhibited by them, and they demonstrated cytotoxicity against A549 lung cancer cells (IC_50_ = 10.46 µg/mL). Dimethylsulfoxonium formylmethylide, a significant bioactive chemical, showed a moderate affinity for interacting with *Salmonella typhi* and *Staphylococcus aureus* pathogenic proteins, according to the molecular docking study. It is appropriate for oral drug formulations due to its low lipophilicity, high gastrointestinal absorption, and few drug–drug interactions, as proven by ADME research.

## 1. Introduction

Currently, the most significant and quickly developing area of research is nanotechnology [1]. The applied sciences are greatly impacted by the field of nanotechnology, which is growing at an astonishing rate. This technology can help with the creation and manipulation of nanostructures with dimensions of less than 10 nm. Because of their small size and high surface-area-to-volume ratio, nanoparticles are significant in this technological area, which affects their stability, soluble capacity, electric and physical characteristics, and chemical composition. The peoples of Rome, Greece, Persia, and Egypt employed silver in some capacity to preserve food approximately 5000 years ago [2]. In the past, numerous dynasties utilized the antibacterial properties of silver (Ag) to manufacture and preserve food as well as beverages [3]. Silver-containing systems, particularly silver nanoparticles (AgNPs), have emerged as one of the most promising materials in modern biomedical science. Nowadays, silver nanoparticles are widely accessible and have been explored for a broad range of therapeutic applications, including antimicrobial, anti-inflammatory, and anticancer uses [4,5]. Klasen [6] reported that silver has been utilized as a disinfectant for several millennia. The majority of the silver used to heal wounds in the 1960s was inserted as salts or nano-systems (colloids). Silver nanoparticles (AgNPs) were created by reducing AgNO_3_ with the *Cacumen platycladi* extract, and their antibacterial activity was evaluated. The metal nanoparticles from biological sources have been developed by using a variety of methods available. The AgNPs exhibited superior antibacterial efficacy when tested against *E. coli* and *S. aureus* [7,8]. Low yields have been a problem with physical procedures, while environmental damage has been a problem with chemical methods, since they require toxic solvents and generate dangerous by-products [9]. By reducing or breaking down the salts of metals in solutions with various chemical reducing substances, the chemical approach essentially applies wet chemistry principles to produce the AgNPs [10,11]. On the other hand, chemical operations often involve costly metal compounds, hazardous or noxious solvents, and reductants such as borohydride of sodium and hydrazine. To prevent particle aggregation or to make them compatible with the body’s systems, various types of stabilisers are also required [12,13]. These conventional chemical processes have therefore run into a number of problems, which have spurred scientists to create new methods. In addition, given the potential of AgNPs for biological applications, it is much desired to create straightforward and environmentally friendly methods for their synthesis employing nontoxic chemicals and solvents under ambient settings [14]. The concepts behind “green chemistry” has become quite popular in this context; these ideas centre on substituting chemical products and upgrading or inventing procedures and technologies to drastically cut down or get rid of compounds that might result in damage to both human health and the natural world [15]. Producing NPs from a wet chemical process by [16] is greatly impacted by the green synthesis of NP. It encourages reactions without dangerous stabilizers, reducing agents, or solvents [17]. So far, several environmentally friendly techniques have been used to create AgNPs [18], incorporating microwaves, electrochemical reduction, and supercritical liquid synthesis [19]. In particular, because of their biocompatibility, biomimetic techniques involving proteins, plant extracts, or micro- or marine organisms have gained popularity for the manufacture of AgNPs [20]. Fungi have more promise for NP synthesis than other microbes, according to recent research; in reality, fungi are capable of producing extracellular enzymes that may decrease metal ions and create metal nanoparticles in vast quantities. Additionally, it has been shown that the stabilizing qualities of the produced NPs are caused by the presence of proteins in the fungi’s waste media [21]. Nanoparticles of silver were used in a number of experimental wound treatments. Combining the antibiotic properties of different types of AgNPs with the cell growth properties of other materials could lead to better and more effective wound healing aids in the future [22]. Silver has been utilized for centuries for the treatment and prevention of several kinds of illnesses, with infections being the most prevalent [23]. When lemongrass extract is combined with aqueous gold ions, gold nanotriangles are generated. The NIR absorbance by gold nanotriangles has a substantial impact on the heat exhaustion of cancer cells [24]. Silver nanoparticles possess antifungal, antiviral, antioxidant, antibacterial, antiplatelet action, and anti-inflammatory properties, expanding the range of potential uses for them in industries including cosmetics, coatings, food packaging, and medicine [25,26]. Numerous significant primary and secondary metabolites with additional industrial uses can be synthesized by endophytes. Secondary metabolites are chemical substances with low molecular weight that are produced to enhance the fungus or host plant’s adaptation to diurnal and seasonal fluctuations in the environment. In contrast to metabolites, essential for the expansion and growth of a life form, they are responsible for specific reactions to a variety of abiotic or biotic stimuli. Endophytic association in medicinal plants has been demonstrated in a multitude of studies conducted over the past lengthy period. It is anticipated that these endophytes will produce novel compounds with a diverse array of potential biological applications [27,28]. Most of this study concentrated on a variety of plant and microbial sources for synthesis. *B. ciliata* is a perennial herb of the family Saxifragaceae, commonly known as hairy Bergenia, growing between the altitudes of 800 and 3000 m all over the temperate Himalayas. The plant has been used as a medicine for the treatment of various human diseases for a long time. *B. ciliata* has a long history of usage for diseases in the Himalaya region among many rural communities. The rhizome of *B. ciliata* has been used for curing pulmonary infections, leucorrhea, piles, and for dissolving bladder and kidney stones. It is widely used in the Ayurveda system of medicine for tonic, astringent, antiscorbutic, laxative, spleen enlargement, dysuria, and ulcers. Due to its medicinal properties, it encourages the synthesis of nanoparticles from leaves. *Corynespora smithii*, a lesser-explored fungal endophyte, belongs to a genus known for producing a range of biologically active secondary metabolites, including polyketides and sesquiterpenes with antimicrobial, antioxidant, and cytotoxic properties [29]. Some species of *Corynespora* have also demonstrated the ability to reduce metal ions and stabilize nanoparticles through extracellular enzymatic activity and protein capping [30]. Despite its potential, *C. smithii* has been minimally studied, and its application in green nanotechnology remains underreported.

*B. ciliata*, a high-altitude medicinal herb, has been widely used in traditional systems like Ayurveda and Unani medicine. Recent pharmacological studies have confirmed its antiurolithiatic (stone-dissolving), antioxidant, anti-inflammatory, hepatoprotective, antimicrobial, and anticancer effects, largely attributed to its phenolic acid, bergenin, and arbutin content [31]. These findings reinforce its medicinal value and make it a suitable host for endophytic exploration and biotechnological applications.

The synergistic potential of using *B. ciliata*-derived endophytes for nanoparticle synthesis thus lies in combining traditional knowledge, fungal biochemistry, and modern nanotechnology to create biofunctional materials for therapeutic use.

The current work, in contrast to past ones, focuses on environmentally friendly ways to synthesize AgNPs and presents the synthesis process, characterizations, applications, and projected antibacterial, antioxidant, and anticancer properties in a systematic manner.

## 2. Materials and Methods

### 2.1. Endophytic Fungal Isolation from Bergenia Ciliate Leaves

Plant material, i.e., leaves of *B. ciliata*, was collected from Srinagar, Garhwal, Uttarakhand, India, and identified from BSI, Dehradun (Accession No. 905), India. For the collected plant materials, first, water that was double distilled was used to wash the leaves. Then it was processed for endophytic isolation as per the method described by [28,32], with slight modifications to eliminate dirt and other pollutants. The green leaves of *B. ciliata* were rinsed many times with normal tap water. Under aseptic conditions, the leaf surface was cleaned by submerging it in sterile distilled water for 1 min, 70% ethanol for 1 min, and 2.5% sodium hypochlorite for four minutes, 70% ethanol for 30 s, and a final rinse again with sterilized distilled water three times. The plant leaves were split into 20 segments using a sterile surgical tool. The segments were subsequently meticulously set on PDA, or potato dextrose agar, plate, enriched with 100 mg/L of chloramphenicol to prevent contamination by bacteria, and kept at 28 °C for seven to ten days. The plates were monitored for fungal growth daily, and any hyphae that emerged from the leaf-cutting were cultivated on freshly prepared PDA plates using the hyphal tilting procedure (Appendix A).

### 2.2. Silver Nanoparticles Synthesis Using Fungal Broth

Using a shaker incubator (Scientech Technologies Pvt, Ltd. Indore, Madhya Pradesh, India), the isolated fungi from *B. ciliate*, a medicinal plant, were mass cultivated in a mycological broth (HiMedia Laboratories Pvt Ltd., Mumbai, India) for 25 days at 25 °C ± 1.5 with constant shaking at 150 rotation per minute (RPM). Following fermentation, Whatman filter paper No. 42 was used to filter the broth, then through a membrane filter assembly with a 0.45 µm size under sterile conditions. The resulting filtrate was utilized to synthesize silver nanoparticles by mixing 10 mL of fungal extract with 90 mL of 1 mM aqueous silver nitrate (AgNO_3_) solution in a 250 mL Erlenmeyer flask. The reaction mixture was then heated at 60 ± 2 °C for 5–8 min with constant stirring to initiate nanoparticle formation. After this heating step, the mixture was cooled to room temperature and incubated in the dark for 24 h to complete the reduction of silver ions. A change in colour to brown indicated the successful synthesis of silver nanoparticles (Figure 1). After incubation in the dark for 24 h, the formation of silver nanoparticles was confirmed by the change in colour to dark brown. The nanoparticles were then separated from the reaction mixture by centrifugation at 12,000 rpm for 15 min at 4 °C. The resulting pellet was washed three times—first with sterile distilled water and then with 70% ethanol—to remove any unbound biological material and residual reactants. Finally, the purified nanoparticles were dried in an oven at 50 °C for 12 h, and stored in airtight vials for further characterization and biological assays.

### 2.3. Characterization of Silver Nanoparticle

To characterize the AgNPs, multiple techniques were used for the characterization to know that the synthesized nanoparticles are within the range of nano scale.

#### 2.3.1. UV Spectroscopy

The phyto-synthesized nanoparticles were exposed to a scanning range between 200 nm and 800 nm to determine the maximum absorbance at which the silver nanoparticles were produced. Using a double beam UV-Vis spectrophotometer, the UV-Vis-spectrum of the produced AgNPs was recorded (Spectronic Model AU2700, Ambala, India).

#### 2.3.2. XRD (X-Ray Diffraction) Analysis

Using a PAN analysis instrument (Xpert Pro Model, Hengelo, Netherlands), fungal-assisted nanoparticles (FANPs) were analysed using X-ray diffraction. The instrument operated at 45 kV and 40 mA, with Cu Kα irradiation within a θ–2θ angle range of 0 to 80°. Scherrer’s equation [33] was employed to estimate the average size of the FANP crystals, and the optical absorbance was computed throughout a wavelength range of 200–800 nm.D=0.9λβ12cosθ

The variables D, λ, β_1/2_, and θ denote the crystallite’s size, X-ray wavelength (CuKα), full width at half maximum (FWHM), and diffraction angle, respectively.

#### 2.3.3. FTIR (Fourier Transform Infrared) Analysis

FT-IR spectra of the silver nanoparticles biosynthesized by *C. smithii*, silver nitrate solution, and washed silver nanoparticles were measured to be between 400 and 500 cm^−1^ (Varian 3100, Church Stretton, Shropshire, UK) using an FTIR spectrometer to confirm whether the organic compounds are free in solution, surface-bonded to the AgNPs, or trapped inside the AgNPs. A Nicolet 6700 FT-IR spectrometer with a 4 cm^−1^ resolution attachment was employed to record the FTIR spectra of a biosynthesized silver product. At a resolution of 4 cm^−1^, all measurements were conducted within the 400–4000 cm^−1^ range.

#### 2.3.4. SEM with EDX Analysis

The morphology and elemental composition of the synthesized silver nanoparticles were analysed using scanning electron microscopy (SEM) coupled with energy-dispersive X-ray spectroscopy (EDX). SEM analysis was performed on a JEOL JSM-5600, Chicago, USA system at an acceleration voltage of 15 kV, with a working distance of 10 mm, and images were captured at magnifications ranging from 5000× to 50,000×, depending on particle visibility and resolution requirements.

For SEM sample preparation, approximately 20–50 µL of the AgNP suspension was carefully dropped onto a carbon-coated copper grid and allowed to dry under a mercury lamp for 4–5 min. Excess sample was removed using filter paper. The dried grid was then loaded into the SEM chamber for imaging [34].

Elemental analysis of the silver nanoparticles was conducted using EDX spectroscopy attached to the same SEM unit. The EDX spectrum confirmed the presence of elemental silver with a strong signal peak around 3 keV, characteristic of metallic silver, along with minor peaks for carbon and oxygen attributed to the capping agents and biological matrix from the fungal extract.

#### 2.3.5. TEM-Analysis

HR-TEM was used to investigate the size and shape of the produced NPs. A very diluted mixture of nanoparticles was dropped onto a copper grid covered with carbon to create samples. The samples were placed in a specimen container and allowed to dry by evaporating at room temperature prior to being placed in the desiccators. The TEM measurements were performed at a 200 kV voltage using a HRTEM (Jeol/JEM 2100, Chicago, IL, USA) with a W-source and a resolution of 0.14 nm, equipped with an ultrahigh resolution pole piece.

#### 2.3.6. Dynamic Light Scattering (DLS) Analysis and Zeta Potential Analysis

Dynamic light scattering (DLS) is one method to utilize when determining the size of nanoparticles in a solution is necessary. The size and form of the nanomaterials in the solution dictate the direction of this kind of monochromatic light scattering [35]. A Zetasizer Nano ZS from Malvern Instruments Ltd., Worcestershire, UK was used to monitor the dynamic changes in light scattering intensity brought on by the particles’ Brownian motion in order to ascertain the pattern of particle sizes. The evaluation yielded the following results: the index of polydispersity (PdI), which showed the extent of the particle size distribution; the median hydrodynamic dimension of each particle; and the highest possible values of the hydrodynamic distribution’s diameter. A value of 0 indicates monodispersity, while a value of 1 indicates polydispersity. The PdI scale is characterized by values ranging from 0 to 1. At a temperature of 25 °C, each measurement was conducted at least three times, and each cycle was allowed to acclimate to the temperature. The data processing procedure employed was high multi-modal resolution. After the AgNP suspension was diluted to concentrations of 20%, 40%, 60%, 80%, and 100%, it was passed through a 0.22 μm syringe-driven filter to remove any large particulates. All dynamic light scattering (DLS) measurements were performed on the filtered samples to ensure an accurate assessment of nanoparticle size distribution. In order to assess the integrity of nanoparticles throughout the cytotoxicity studies, nanoparticles were subjected to DLS analysis at various time points: at the time of treatment (0 h), 24 h post-treatment, and at the endpoint (48 h) of the experiment before the addition of the MTT reagent.

Zeta potential measurements were performed using the same Zetasizer Nano ZS (Malvern Instruments Ltd., Worcestershire, UK) used for DLS, at 25 °C. The samples were diluted with Milli-Q water and filtered using a 0.22 μm syringe filter before analysis. The electrophoretic mobility of the particles was measured using laser Doppler velocimetry, and zeta potential values were calculated using the Smoluchowski equation. Each measurement was conducted in triplicate to ensure accuracy and reproducibility. The zeta potential provided insight into the surface charge and colloidal stability of the synthesized silver nanoparticles.

#### 2.3.7. GC–MS (Gas Chromatography–Mass Spectrometry) Analysis of Synthesized AgNPs

The GC–MS analysis was conducted to identify phytochemical constituents in the methanolic extract of *B. ciliata*. It is important to note that no silver nanoparticles were involved in the GC–MS analysis, and their dissolution in methanol was not part of this procedure.

Approximately 1 μL of the methanolic extract was injected using a micro-syringe into the GC–MS system (Shimadzu, Model: GCMS-TQ8040, Buckinghamshire, UK) equipped with an SH-Rxi-5Sil MS column (30 m length × 0.25 mm ID × 0.25 µm film thickness), which comprises 5% diphenyl and 95% dimethylpolysiloxane. The injector temperature was maintained at 250 °C with a split-mode injection. The oven temperature was initially set at 60 °C for 15 min, then increased to 280 °C at a ramp of 10 °C/min and held constant to elute higher molecular weight compounds.

Helium was used as the carrier gas at a constant flow rate of 1.0 mL/min, not methanol, which was only used during the extraction phase. Compound identification was achieved by comparing the mass spectra with entries in the NIST library.

### 2.4. Antibacterial Activity of Synthesized AgNPs

Four Gram-positive bacteria (*B. subtilis*, *Staphylococcus aureus*, *Bacillus cereus*, and *Streptococcus pneumoniae*) were used to assess the antibacterial efficacy of fungal-assisted produced AgNPs, and four Gram-negative bacteria (*Salmonella typhi*, *Klebsiella pneumoniae*, *Escherichia coli*, and *Pseudomonas aeruginosa*). The procedure described by [36] was implemented to evaluate the antibacterial efficacy of the FANPs. After the solidification of Mueller–Hinton agar (MHA), we distributed 10^8^ cfu/mL of each bacterial species over the surface [36]. Subsequently, the mixture was desiccated. Then, we created a 6-mm well for the AgNPs and a negative control (DMSO). Gentamicin and chloramphenicol were employed as positive controls. A 100 mg/mL FANP solution in DMSO (Dimethyl Sulfoxide) was administered to the wells, while 100 µL of DMSO was used as the negative control. For 18 to 24 h, at 37 °C, the plates were incubated upright. The zone of inhibition was calculated and documented after the period of incubation in compliance with Chandra et al. [37].

### 2.5. Molecular Docking of the Major Component of the FANPs Against Pathogenic Bacteria

The bioactive primary component of the FANPs, dimethylsulfoxonium formylmethylide, was chosen for molecular docking with *S. aureus* and *S. typhi*, two pathogenic microbes that cause illnesses in humans. A GC–MS analysis was used to gather this data. Protein–ligand interaction was studied using the CB-dock-2 online tool [38]. The crystal structure of dimethylsulfoxonium formylmethylide was found using the PubChem database [https://pubchem.ncbi.nlm.nih.gov/ accessed on 21 December 2024]. The Protein Data Bank (PDB, https://www.rcsb.org/ accessed on 21 December 2024) provided the protein structures of the 5tw8 (penicillin-binding protein 4), 1mwr (penicillin-binding protein 2a from methicillin-resistant *S. aureus* strain 27r) of *S. aureus*, 5ztj (gyrase A C-terminal domain), and 3uu2 (OmpC is an outer membrane porin) from *S. typhi*, 2zfg (outer membrane porin, OmpF) of *Escherichia coli*, 4ng2 (LasR LBD–QslA complex) of Pseudomonas. The PDB file of pathogenic bacteria, which contained a protein molecule, was docked with the SDF file of a ligand (DMSF) molecule.

### 2.6. ADME (Absorption, Distribution, Metabolism, and Excretion) Properties and Drug-Likeness Prediction of DMSF

The drug-likeness prediction and models of ADME of the ligands were investigated using Swish ADME, a free web application (accessed on 21 December 2024). Lipinski’s rule of five (L2F) was used to examine the molecule’s drug-likeness in which there is the following rule to be followed, i.e., molecular mass less than 500 Dalton, high lipophilicity (expressed as LogP less than 5), less than 5 hydrogen bond donors, less than 10 hydrogen bond acceptors, and molar refractivity should be between 40 and 130. In other words, there are ten hydrogen bond acceptors and five hydrogen bond donors. The medication has a molecular weight of less than 500 Da. A computed log *p* ≤ 5 and polar surface area (PSA) ≤ 140 Å2. The test molecule is deemed impermeable or not orally bioavailable if two or more of these conditions are not satisfied [39].

### 2.7. Antioxidant Activity of Synthesized FANPs

A DPPH (1, 1-diphenyl-2-picrylhydrazyl) test was used to assess the antioxidant capability of the produced AgNPs. Utilizing a slightly modified version of [40], we used DPPH, a stable synthetic free radical, to test the compounds’ ability to donate radical hydrogen or scavenge free radicals. After combining the extract with different concentrations of 0.002% DPPH solution (10, 50, 100, 150, 200, 250, 500, and 1000 μg/mL), it was left to stand at room temperature for half an hour. Using a twin-beam UV-visible spectrophotometer, the absorbance at 517 nm was measured 30 min later. The DPPH radical’s scavenging is directly correlated with the colour shift from pink to yellow.RSC (%) = [(A_C_– A_S_)/A_C_] × 100

This is where the absorbance of the DPPH methanolic solution is represented by A_C_, and the absorbance of the sample containing silver nanoparticles is represented by A_S_. We determined the samples’ mean RSC (in %) for each extract concentration. The FANP concentration that gave 50% inhibition EC_50_, or the concentration at which 50% of DPPH was effectively scavenged, was found by interpolating from linear regression analysis. Ascorbic acid was used as the standard antioxidant agent.

### 2.8. Antiproliferative Analysis of Synthesized FANPs

#### 2.8.1. Culturing of Cancer Cell Line

For this study, the human non-small cell lung cancer line A549 was obtained from the National Centre for Cell Sciences in Pune, India. In a humidified cell culture incubator set at 37 °C with 5% CO_2_, the A549 cells were cultivated in Roswell Park Memorial Institute (RPMI-1640) media supplemented with 10% foetal bovine serum (FBS) and 1% streptomycin/penicillin/amphotericin B solution. Every other day, the culture media was replaced, and cells were passaged till it attained 70–80% confluent growth. The cells were trypsinized and collected after attaining confluence, and they were centrifuged for five minutes at 1000–1200 rpm. The cell particle was then mixed with a cooling medium containing 10% DMSO, 20% FBS, and 70% full RPMI-1640 media. A portion of 1 × 10^6^ cells in 1 mL of suspension was placed in cryovials. The cryovials were subjected to a rapid two-hour transition to −20 °C, and they were subsequently refrigerated to −80 °C for the night. Subsequently, they were deposited in the presence of liquid nitrogen for long-term preservation.

#### 2.8.2. MTT Assay

Making use of 3-(4,5-dimethylthiazolyl-2)-2,5-diphenyltetrazolium bromide (MTT), a tetrazolium chemical, the MTT assay was used to determine the vitality of the cells. With a few minor adjustments, the process was carried out in accordance with the description in [41]. Cells were grown in RPMI-1640 media for 24 h, after being seeded in 96-well culture plates with 5000 A549 cells per well. At this juncture, various concentrations of nanoparticles were introduced in triplicate and incubated for a duration of 48 h prior to the addition of MTT reagent for the evaluation of cytotoxicity. The experiment was repeated for at least three times, three wells per concentration at each experimental repeat. Results were recorded to estimate the IC_50_ value, and the statistical significance was estimated. Vinblastine, an anticancer drug, was administered for 48 h as a positive control. Vinblastine was utilized at working concentrations of 2, 5, 10, 20, 50, and 100 μg/mL, with a stock concentration of 90 mg/mL. Following a brief pre-mixing step, the plates were treated and then read at 570 nm using a micro-plate reader. Calculations of cell viability percentages were performed according to [41]. Three separate trials, each with six replicates, provided the data.

## 3. Results and Discussion

Silver nanoparticles (AgNPs) are essential components of nanomedicine and nanotechnology. The nanoparticles of silver are the focus of chemical and biological studies by scientists, as they are an exceptional weapon against a variety of hazardous microorganisms [42]. Experiments by [43,44] have shown that fungi may produce spherical silver nanoparticles, has and have determined that Ag+ ions may be reduced by certain microorganisms. Various microbes and their metabolic metabolites, which serve as stabilizing and reducing agents, may facilitate the biogenic synthesis of nanoparticles [45]. To enhance durability and possibly discover biological activity, these nanoparticles are encapsulated by compounds from the organisms that are used to manufacture them. tThe biocompatibility of biogenic materials in nanoparticle applications can be enhanced by their relatively simple, clean, economical, sustainable, and biocompatible synthesis [46,47].

### 3.1. UV Spectrophotometer Analysis

UV-Vis spectroscopy is a straightforward, efficient, and user-friendly characterization instrument that is capable of measuring optical properties, stability, and synthesis process parameters, like time, temperature, and pH. In addition to the AgNPs sample, a blank control spectrum of the fungal extract in Milli-Q water (without AgNO_3_) was also recorded under identical conditions. This control was used to subtract the background absorbance contributed by the biomolecules in the fungal filtrate. The absence of a surface plasmon resonance (SPR) peak in the blank spectrum confirmed that the characteristic peak observed at ~418 nm in the AgNPs spectrum was due to nanoparticle formation. The surface plasmon resonance (SPR) phenomenon induces charged surface production and free electron oscillations in nanoparticles when they are exposed to electromagnetic radiation [48]. The characteristic peak observed at ~418 nm corresponds to the surface plasmon resonance (SPR) of the silver nanoparticles. The SPR is a collective oscillation of electrons on the nanoparticle surface when excited by light, resulting in a strong absorbance in the visible region. This optical property is unique to metallic nanoparticles and confirms their successful synthesis and nanoscale size. The development of silver nanoparticles is indicated by a change in pigment from clear to brown. The fungal-mediated silver nanoparticles’ greatest absorbance was observed at 418 nm (Appendix A), when the UV-Vis spectrophotometer was employed to scan the wavelength between of 200 and 800 nm, thereby confirming the existence of silver nanoparticles. A silver nanoparticle that was generated through a fungal process demonstrated a UV absorbance that ranged to a peak at 420 nm when a UV-Vis spectrophotometer was employed [49,50,51].

### 3.2. XRD (X-Ray Diffraction) Analysis

An x-ray diffraction analysis was conducted to determine the crystalline structure and to estimate the average crystallite size of the synthesized AgNPs. The high-resolution XRD pattern (Figure 2) exhibited prominent diffraction peaks at 2θ values of 38.26°, 46.05°, 64.78°, and 77.53°, which correspond to the (111), (200), (220), and (311) planes of the face-centred cubic (FCC) structure of silver (JCPDS card no. 04-0783). The crystalline size of the FANPs was determined using Scherrer’s equation. Using the most intense (111) peak, the average crystallite size was calculated to be approximately 18.6 nm. The observed peaks and their glancing angles are in excellent agreement with previous results, as evidenced by the appearance of cubic-structured crystalline silver nanoparticles [52]. The XRD result is consistent with previous findings that were derived from plant-based synthetics [53].

This result aligns closely with the TEM measurements (11.0–20.4 nm) and supports the presence of nanocrystalline silver particles. Minor variations between XRD and TEM results can be attributed to the fact that XRD measures the size of coherent diffracting domains (crystallites), while TEM includes the total particle size, which may include amorphous biological capping layers.

### 3.3. FTIR Analysis of Synthesized Nanoparticle, Silver Nitrate, and Washed Silver Nanoparticle

The FTIR spectrum of the synthesized silver nanoparticles (AgNPs) showed notable absorption bands at the following regions. At ~1636 cm^−1^, this peak corresponds to C=O stretching vibrations typically found in amide or carboxylic acid groups. Its presence suggests that proteinaceous or organic molecules—possibly from the fungal extract—are adsorbed or bonded onto the nanoparticle surface, acting as capping agents. In the 3200–3900 cm^−1^ region (notably at ~3391, 3452, 3547, and 3870 cm^−1^) (Figure 3), intense peaks indicate O–H and N–H stretching vibrations, which are characteristics of hydroxyl and amine groups from phenolic compounds, alcohols, or amide-containing proteins. This again implies that fungal biomolecules are interacting with the nanoparticle surface. In the 500–600 cm^−1^ region (e.g., peaks at 502, 531 cm^−1^), peaks may correspond to Ag–O or Ag–N stretching, confirming that silver is forming bonds with organic functional groups, indicating surface functionalization. The FTIR results demonstrate the presence of organic functional groups—carbonyl (C=O), hydroxyl (O–H), and amine (N–H)—on the silver nanoparticles. This strongly supports that the GC–MS-identified compounds (e.g., dimethylsulfoxonium formylmethylide, hexadecanoic acid) or related biomolecules from the fungal extract are not freely floating in solution but are surface-bound or involved in capping/stabilizing the AgNPs. This provides evidence that the observed biological activity is likely due to the synergistic action of AgNPs with these surface-bound biomolecules, thus validating their relevance to the nanoparticle formulation.

### 3.4. SEM with EDX

Scanning electron microscopy was performed for the determination of the shape and arrangement of the synthesized FANPs. Figure 4 shows the SEM analysis of the FANPs in which the nanoparticles appear to be irregular in shape.

This SEM image captures the microstructural details of silver nanoparticles synthesized using fungal mediation, illustrating the surface morphology and distribution of the nanoparticles. The image was obtained at a magnification that allows for a scale bar indicating 40 µm, providing a comprehensive view of the observed features. The sample depicted in the image consists of silver nanoparticles produced through the mediation of fungi. When it comes to producing nanoparticles, this biosynthetic process is well-known for being both economical and environmentally beneficial. The magnification level is such that the scale bar represents 40 µm, allowing for precise measurement of the nanoparticles and their distribution across the sample. The image reveals a heterogeneous distribution of silver nanoparticles. The nanoparticles appear irregular in shape and are dispersed across the fungal matrix. The surface morphology indicates a successful synthesis of nanoparticles, with some areas showing agglomeration. Detailed observation shows that the nanoparticles are embedded within the fungal matrix, suggesting effective interaction between the fungal biomass and silver ions during the synthesis process. The size distribution of the nanoparticles varies; the average size of the synthesized FANPs was 40.16 nm. The findings presented in this SEM image are consistent with other studies on fungal-mediated silver nanoparticles [54]. For example, in [44], it was shown that *Fusarium oxysporum* could biosynthesize silver nanoparticles, showing similar irregularly shaped nanoparticles embedded within the fungal matrix. The use of *Aspergillus fumigatus* in the formation of silver nanoparticles was also documented by [55], where SEM images revealed a comparable distribution and morphology of the nanoparticles. Nanoparticles in the same size range as those discovered in our work were biosynthesized utilizing *Pseudomonas stutzeri* in a study conducted by [56], confirming the effectiveness of microbial mediation in nanoparticle synthesis. The contrast and composition variations observed in our SEM image align with findings by [56], where differences in the density and distribution of nanoparticles were highlighted using SEM analysis.

The prominent peaks in the EDX spectrum correspond to various elements present in the sample. The most significant peak, at approximately 3 keV, is attributed to silver, confirming the successful synthesis of silver nanoparticles. Peaks around 0.3 keV indicate the presence of carbon, likely originating from the fungal matrix used in the synthesis process. A peak near 0.5 keV suggests the presence of oxygen, which may be associated with organic components of the fungal biomass or oxide formation on the nanoparticle’s surfaces (Appendix A). The intensity of the peaks correlates with the relative abundance of the elements in the sample. The prominent silver peak confirms its dominant presence, which is expected as silver nanoparticles are the main focus of this study. The carbon and oxygen peaks, though significant, are less intense, indicating their presence in the fungal matrix rather than as primary constituents. The EDX spectrum provides critical insights into the elemental composition of fungal-assisted silver nanoparticles. The confirmation of silver as the predominant element validates the effectiveness of the biosynthesis process. Oxygen and carbon components help to support the theory that fungal biomass is involved, which plays a crucial role in the synthesis and stabilization of the nanoparticles. The trace elements identified, such as chlorine and molybdenum, could be further investigated to understand their origins and potential impacts on the nanoparticle properties. This analysis aligns with previous studies, such as those by [54], which reported similar elemental compositions in biosynthesized silver nanoparticles. The consistency in elemental detection across different studies enhances the credibility of the fungal-mediated synthesis method for producing silver nanoparticles with desired characteristics and purity.

### 3.5. Transmission Electron Microscopy (TEM) Analysis of FANPs

The TEM study revealed that the silver NPs appear as distinct, spherical particles scattered across the field of view. The distribution seems relatively uniform, with some areas showing clusters of particles, indicating potential aggregation. The particles vary in size, with measurements labelled on the image. The sizes range between 11.0 and 20.4 nm in diameter (Appendix A). This indicates that the fungal synthesis process produced nanoparticles within the nanoscale range, with some variation in size. The particles appear darker against a lighter background, which is typical for TEM images, as the electron-dense silver scatters electrons more effectively. The image has good contrast, allowing for clear identification and measurement of individual nanoparticles. The scale bar at the bottom right indicates 20 nm, providing context for the size of the nanoparticles. The image is taken at a high magnification of 500,000×, allowing for detailed observation of the nanoparticle’s morphology. The background suggests that the nanoparticles are dispersed in a possibly amorphous or less electron-dense medium, which could be a residual biological matrix from the fungal synthesis process. These results are consistent with earlier research that documented using a variety of fungal species to produce silver nanoparticles, typically resulting in nanoparticles within a similar size range. For instance, studies by [54,57] reported the development of AgNPs between 10 and 50 nm in size when synthesized using *Aspergillus niger* and *Penicillium* sp., respectively. The nearly spherical morphology of the nanoparticles observed in the TEM images is consistent with previous reports on biologically synthesized AgNPs. This morphology is commonly attributed to the reduction of silver ions by fungal biomolecules, including enzymes, proteins, and secondary metabolites, which not only facilitate the reduction process but stabilize the formed nanoparticles. The presence of such biomolecules is known to influence the shape and size of nanoparticles, as evidenced by [44], who reported similar spherical shapes in AgNPs synthesized using *Fusarium oxysporum*. The relatively narrow size distribution observed in the TEM images suggests a controlled synthesis process, which can be attributed to the uniformity of the reducing agents present in the fungal extract. This is crucial, as the size of nanoparticles significantly impacts their physical, chemical, and biological properties.Smaller nanoparticles, such as those observed in this study, often have greater surface-area-to-volume ratios, which strengthen their antibacterial properties, as demonstrated in the work of [58]. Furthermore, the slight aggregation of nanoparticles observed in some regions of the TEM image might be indicative of inadequate capping by fungal metabolites or the presence of ionic interactions between particles. This phenomenon has been reported in other studies where biologically synthesized nanoparticles showed a tendency to form clusters due to partial stabilization by organic molecules from the fungal extract [59]. While aggregation may influence the overall efficacy of the nanoparticles, it also sheds light on the kind of capping agents that the fungus uses while synthesizing. Due to its environmentally benign nature and the capacity of fungi to produce a variety of bioactive metabolites that can stabilize the nanoparticles, fungal-mediated synthesis of silver nanoparticles has drawn a lot of attention. Recent studies have shown that a variety of fungi, including the Aspergillus species, are effective in producing silver nanoparticles that possess strong antibacterial and cytotoxic qualities. For example, it has been demonstrated that *A. flavus* can produce silver nanoparticles with diameters comparable to those found in our study with substantial antibacterial action [60]. Furthermore, the TEM picture demonstrates how much smaller the NPs are, which is helpful for biomedical applications. Studies demonstrating that NPs in the 10–20 nm range are very efficient in piercing cell membranes and causing cytotoxic effects have supported this property [61].

### 3.6. Dynamic Light Scattering (DLS)

Dynamic light scattering (DLS) can be employed to measure the diameters of nanoparticles that are dispersed in a liquid. DLS regulates particle size and dispersion in physiological solutions. Appendix A shows the AgNPs’ dynamic light scattering (DLS) histogram, which shows an average particle diameter of 100 nm. The TEM diameters and the XRD values were both substantially smaller than the DLS sizes. While DLS measures the hydrodynamic diameter, TEM measures the physical size distribution, which does not account for any capping agent. According to [62], the hydrodynamic diameter of a particle comprises its diameter as well as any ions or molecules that are attached to its surface and circulate with the AgNPs in solution. This might be the cause of the disparity. The comparative particle size from different characterization techniques is shown in Table 1.

As depicted in the figure, the photosynthesized silver nanoparticle solution, which was prepared from 10 mL of fungal extract, exhibited a zeta potential value of −17.4 mV, indicating that the generated nanoparticles are repellent (Appendix A).

### 3.7. GC-MS Analysis of FANPs

The technique of gas chromatography was used to separate the chemicals by their mass spectrometry profiles. Retention times, base *m*/*z* values, and peak areas were recorded for each detected compound. Biological activities were referenced from the existing literature based on the identified compounds.

The GC–MS chromatogram showed dimethylsulfoxonium formylmethylide as the major compound in the sample (Appendix A), which is associated with its biological activity, as documented in the literature. This compound has been recognized for its significant antimicrobial, antioxidant, and anticancer properties, underscoring its potential utility in medical and pharmaceutical applications [63]. The high concentration of dimethylsulfoxonium formylmethylide in the sample aligns with its documented efficacy, reinforcing its importance as a focal point for further research. Additionally, the peak of hexadecanoic acid and 9-octadecenoic acid, although present in lower amounts, contributes further therapeutic potential (Table 2). Hexadecanoic acid has been associated with anti-inflammatory and analgesic effects, while 9-octadecenoic acid is known for its lipid-lowering and antioxidant properties [64]. These substances are worthy of further investigation, since their slight presence raises the possibility that they work in concert to improve the sample’s overall biological activity. Differences in the chemical characteristics and interactions of these substances within the GC column are highlighted by the changes in retention periods and *m*/*z* values that were detected during the GC–MS analysis. These variations can affect how the chemicals behave in biological systems and provide light on their molecular properties.

Understanding these variations is crucial for optimizing the extraction and purification processes, and for developing effective therapeutic strategies. Overall, the findings from this GC–MS analysis offer a robust basis for exploring the therapeutic applications of dimethylsulfoxonium formylmethylide, hexadecanoic acid, and 9-octadecenoic acid. Future research should focus on elucidating the mechanisms underlying their biological activities and assessing their potential in clinical settings [70,71]. These insights will be instrumental in advancing the development of new treatments and enhancing our understanding of the therapeutic potential of these compounds. The differences in retention times and *m*/*z* values indicate variations in the compounds’ chemical properties and interactions within the GC column.

### 3.8. Antibacterial Activity of Synthesized Silver NPs

The antibacterial properties of the nanoparticles were evaluated in a variety of bacteria, including four Gram-negative bacteria and four Gram-positive bacteria. See Table 3 for the antibacterial efficacy of eight bacterial species against the fungal-assisted silver nanoparticles.

No bacterial species could tolerate the inhibitory effects of the generated silver nanoparticles. The highest activity was shown by *S. aureus* (i.e., 20.85 ± 1.20 mm), followed by *B. subtilis* (i.e., 18.95 ± 0.35 mm), and the least activity was shown by *B. cereus* (i.e., 15.0 ± 0.14 mm) (Appendix A). The synthesized FANPs from *Aspergillus terreus* were also reported to have antimicrobial activity against *S. aureus*, *P. aeruginosa*, and *E. coli*, which could prevent a wide range of harmful organisms, such as fungi and bacteria, with remarkable efficiency [72,73].

Antibiotic resistance among bacterial species is one of the global problems faced by researchers [70,74], and it is very difficult to reverse the resistance in bacteria; there is little in the literature and few research articles available that claim the development of new effective antibiotics. The nanoparticles synthesized through greener technology have shown some hope in controlling the growth of pathogenic bacteria. The fungal-mediated nanoparticles have shown promising results against all tested bacteria, and it can be a better option for the treatment of pathogenic bacteria. *Salmonella typhi*, *Escherichia coli*, *Shigella dysenteriae*, and *Listeria monocytogenes* were among the species examined that showed antibacterial capacity against the AgNPs produced by the endophytic fungus *Talaromyces purpureogenus*, which was isolated from *T. baccata* Linn [73].

### 3.9. Molecular Docking of DMSF with Proteins of Pathogenic Bacteria

The ligand (dimethylsulfoxonium formylmethylide) was docked with the proteins *S. aureus* (1mwr and 5tw8) and *S. typhi* (5ztj and 3UU2) using the CB-dock online tool-2, which aids in predicting the ideal binding location of the ligand with the protein. The results of the CB-dock online tool-2 examine all evident molecular interactions that influence the activity of the protein–ligand assembly complex, and the binding energy needed for its development (Figure 5). Dimethylsulfoxonium formylmethylide was discovered to have a moderate binding affinity of −4.0 and −3.8 kcal/mol for the pathogenic protein 5tw8 and 1mwr of *S. aureus*, respectively (Table 3). Similarly, the moderate binding affinity of dimethylsulfoxonium formylmethylide was determined to be −3.9 and −3.5, kcal/mol when it was docked with the 3uu2 and 5ztj protein of *S. typhi*, and binding energy −3.8, and −3.8 kcal/mol was observed when docked with the 2zfg, and 4ng2 protein of *E. coli* and Pseudomonas, respectively (Table 4).

### 3.10. ADME and Drug-likeness Prediction of Active Molecule i.e., Dimethylsulfoxonium Formylmethylide

DMSF has a low polar surface area (TPSA: 42.52 Å^2^), a low conformational flexibility (one rotatable bond), and a simple molecular structure (MW: 120.17 g/mol, seven heavy atoms). These features imply that it is unlikely that there are substantial steric or electrical obstacles to the transport of DMSF though biological membranes. The presence of two hydrogen-bond acceptors and the absence of hydrogen-bond donors suggests efficient passive nonpolar diffusion between the membranes associated with selective pan-cancer molecular interaction, as shown by [74]. DMSF shows low lipophilicity (Consensus Log Po/w: −0.11), implying minimal interaction with lipid rich environments, reducing bioaccumulation and off-target effects. It is a top candidate for aqueous formulations due to its incredibly high solubility across prediction models (e.g., ESOL: 100 mg/mL; SILICOS-IT: 28.8 mg/mL) (Appendix A). ITS hydrophilic characteristics enhance its oral administration suitability and possibly help its pharmacokinetic balance in the bloodstream. The gastrointestinal absorption of the compound is therefore an important consideration for candidates of oral medication. Targeting peripheral systems is advantageous, however, as compounds with limited BBB penetration avoid side-effects related to the central nervous system. In addition, the limited drug–drug interaction potential is augmented by lack of interactions with P-glycoprotein (P-gp) and the major cytochrome P450 enzymes (CYP1A2, CYP2C19, CYP2C9, CYP2D6, CYP3A4). PAE has limited delivery routes due to the high Log Kp value (−7.53 cm/s), showing low skin efficacy. DMSF adherence to Lipinski’s Rule of Five (violations: 0) supports the oral bioavailable drug potential of DMSF. But the fact that it fails the Ghose and the Muegge filters raises some concerns with respect to the low molecular weight (175 Da) and the lower count (14) of atoms, which affect the receptor targeting efficiency and specificity. Despite the last two pitfalls, the compound has an acceptable bioavailability score, a moderate value (0.55), while also obeying the Veber and Egan filter, indicating good characteristics of absorption and transport in in vivo conditions (Appendix A).

Two Brenk alerts (aldehyde, thiocarbonyl group) and one PAINS alert (thioaldehyde) are identified by a medicinal chemistry study, all raising potential stability and reactivity issues. These functional groups can render the molecule more prone to in vivo metabolism liability and off-target effects; therefore, careful optimization should be performed during the leads’ development. Nonetheless, its synthetic accessibility score (2.61) suggests that DMSF is relatively simple to synthesize, making it useful for preclinical and clinical research stages.

The gastrointestinal permeability and brain penetrating effectiveness of DMSF are predicted by the BOILED-Egg plot between the water partition coefficient (WLOGP) and topological polar surface area (TPSA), as illustrated in Appendix A. The molecule represented in the chart demonstrates the properties conducive to drug-likeness, positioning it as a promising candidate for further development. Based on the BOILED-Egg model (Appendix A), the following interpretation can be made about the molecule represented by the red circle. The molecule is located near the centre of the yellow ellipse (BBB region) and within the white ellipse (HIA region). This suggests the molecule has physicochemical properties favourable for both blood–brain barrier (BBB) penetration and high intestinal absorption (HIA). The red circle indicates that the molecule is classified as PGP-, meaning it is not a substrate for P-glycoprotein efflux transporters. This is a favourable property for central nervous system (CNS) drug development, as PGP substrates are often actively pumped out of the brain, reducing effectiveness. The molecule has moderate lipophilicity (WLOGP~1). This indicates it is sufficiently hydrophobic to pass through lipid membranes but not overly hydrophobic, which could lead to issues like poor solubility. The topological polar surface area (TPSA) value (~40) is relatively low, consistent with favourable absorption and the ability to cross the BBB, as lower TPSA generally indicates better permeability. This molecule has an optimal balance of lipophilicity and polar surface area, suggesting it is a strong candidate for oral bioavailability and CNS-targeted drug development. It also avoids PGP-mediated efflux, further enhancing its potential as a brain-penetrating drug.

While the molecular docking of DMSF with selected proteins from *S. aureus* and *S. typhi* revealed moderate binding affinities (–3.5 to –4.0 kcal/mol), this preliminary analysis serves only to indicate potential molecular interactions. These results suggest that DMSF may not exert its antimicrobial effect solely through the inhibition of these specific proteins, and other mechanisms, such as oxidative stress induction, membrane destabilization, or synergy with silver ions or capping biomolecules, should be explored. Future docking studies will expand the analysis to include proteins from other clinically relevant pathogens evaluated in the antibacterial assay, offering a more comprehensive understanding of its bioactivity. While the ADME analysis emphasizes the pharmacokinetic potential of DMSF as a bioactive molecule, it is important to underscore that the silver nanoparticles (AgNPs) synthesized using the green method also play a crucial biomedical role. The AgNPs themselves exhibit potent antibacterial, antioxidant, and anticancer properties, as demonstrated by the in vitro assays. These biological effects may be attributed to the intrinsic antimicrobial properties of silver, enhanced by the capping and stabilization provided by fungal metabolites. Moreover, the green synthesis method contributes to improved biocompatibility and reduces the toxicity risks often associated with chemically synthesized nanoparticles. Together, the AgNPs act as both carriers and active agents, potentially working synergistically with associated bioactive compounds like DMSF to improve therapeutic efficacy.

### 3.11. Antioxidant Activity of Synthesized Fungal Assisted Nanoparticle

DPPH is a simple and rapid method for determining antioxidant activity. A pink colour solution containing DPPH, a stable free radical at low temperatures, can be prepared in methanol. When an antioxidant molecule is present, it undergoes a reduction, which leads to the formation of yellow solutions.

The ability of silver nanoparticles to scavenge free DPPH at ambient humidity was evaluated at dosages that varied between 10 μg/mL and 1000 μg/mL. The concentration of silver nanoparticles increased, resulting in a scavenging action with a dosage response. The maximum free radical scavenging activity was found at the concentration 1000 μg/mL solution of 54.42%. An increase in the RSA of the sample also increases the IC_50_ value of the concentrated sample (Appendix A). The DPPH technique of endophytic fungi revealed that the species Penicillium of *Glycosmis mauritiana* has high antioxidant activity [70]. Recent research has demonstrated that *T. baccata* Linn may contain silver nanoparticles generated by the endophytic fungus *Talaromyces purpureogenus*, which were discovered to possess potent antioxidant properties. The percentage inhibition of myco-synthesized AgNPs was 15.2% at low concentrations and 32.8% at high concentrations. The AgNPs have IC_50_ values of 250.3009 µg/mL [73]. For the synthesis, an endophytic fungal strain called *Pestalotiopsis microspora* that was isolated from *Gymnema sylvestre* leaves were utilized with IC_50_ values of 76.95 ± 2.96 and 94.95 ± 2.18 µg/mL. The AgNPs showed effective radical scavenging activity against DPPH and H_2_O_2_ radicals, respectively, demonstrating their potential as antioxidants [71].

### 3.12. Anticancer Activity

Antiproliferative efficacy against the non-small lung cancer cell line A549 was assessed using the fungus-derived silver nanoparticles. The MTT assay was used to evaluate the survival of the A549 cells under treatment with different doses of the test chemicals. The findings are illustrated in Figure 6. As depicted, the control group (C) demonstrated 100% cell viability. By contrast, cell viability diminished in a dose-dependent manner, resulting in approximately 60%, 55%, 40%, and 15% viability following treatment with nanoparticles at concentrations of 5, 10, 20, and 50 μg/mL, respectively (Figure 6A). Vinblastine was served as an anticancer control that also showed cytotoxicity as a dose dependent activity (Figure 6C). The dose–response curve was fitted using a logistic model to determine the IC_50_ value, which was calculated to be 19.21 μg/mL (Figure 6B), while for Vinblastine it was 1.5 μM/mL (Figure 6D). This value denotes the concentration of the tested compound that is sufficient to prevent 50% of the cell viability, indicating its potency in reducing A549 cell proliferation. These findings are consistent with previously reported results using the MTT assay to assess cell viability and cytotoxicity [75,76].

The impact of FANPs on cell viability at different concentrations (5, 10, 20, and 50 µg/mL) in comparison to the control group (C) is depicted in the bar graph. The y-axis shows the percentage of cells that are viable, and the x-axis shows the various FANP concentrations. At 5 µg/mL, cell viability drops to between 60 and 70 percent, suggesting that FANPs start to show anticancer activity at this dose by lowering cell viability by 30 to 40 percent. Cell viability is further decreased to around 50–60% at 10 µg/mL, showing a greater anticancer activity with a 40–50% drop in cell viability. At 20 µg/mL, the cell viability decreases to about 40–50%, indicating a >50% decrease in cell viability and an increased anticancer activity. At 50 µg/mL, cell vitality dramatically dropped to about 20–30% at the highest concentration tested, suggesting the most effective anticancer activity with a reduction in cell viability of 70–80%. The FANPs show dose-dependent anticancer action, as the graph illustrates (Figure 6).

The IC_50_ value of the FANPs against the lung cancer cell line is 19.21 µg/mL, while for the positive control Vinblastine it was 1.5 μM/mL. Cell viability declines with increasing FANPs concentration, suggesting greater anticancer activity at higher doses. Among the examined doses, 50 µg/mL shows the greatest reduction in cell viability, indicating that this concentration has the most anticancer activity. This study shows that FANPs have the capacity to be a cancer prevention medication, which calls for more research into their mode of action and effectiveness in other cancer cell lines. To ascertain that the nanoparticles during the treatment window of 48 h preserved their integrity and did not form substantial aggregates, we conducted DLS and TEM analyses at the commencement of treatment (0 h), after 24 h of treatment (24 h), and immediately prior to the cessation of anticancer treatment (48 h). The results, illustrated in Figure 7, demonstrated that the nanoparticles maintained their size and exhibited no aggregation throughout the incubation period (0 h, 24 h, and 48 h DLS).

The data presented in Figure 7A,B indicate that the nanoparticle size at the initiation of treatment was 53.28 nm according to DLS, while TEM revealed a well-separated average size of 15 nm. After 24 h, there was minimal variation, with an average size of 61.95 nm by DLS and 15 nm by TEM, as shown in Figure 7C,D. The integrity of the nanoparticles remained intact following 48 h of treatment, coinciding with the point at which cytotoxicity treatment was halted; the average size recorded by DLS was 98.56 nm, while the TEM analysis indicated an average nanoparticle size of approximately 15 nm, as depicted in Figure 7E,F. Collectively, these experiments elucidated a robust antiproliferative activity and suggested that the nanoparticles exhibited considerable stability throughout the treatment duration.

## 4. Conclusions

The globe is facing significant challenges due to the rise of antibiotic resistance among harmful bacteria. The issue of medication resistance is a worldwide one, not just one that affects India. The globe is facing significant challenges due to the rise of antibiotic resistance among harmful bacteria. The research concluded that the fungal endophyte *C. smithii*, which was isolated from *B. ciliate*, has the potential to be used in the production of silver nanoparticles. The produced AgNPs showed potential antioxidant qualities as well as significant antibacterial activity against a variety of harmful bacterial species. This green synthesis approach not only offers a long-term and eco-friendly substitute to traditional methods but opens up new avenues for the development of nanomaterials with enhanced biological activities. The findings underscore the importance of exploring endophytic fungi derived from medicinal plants for biofabricating nanoparticles that may find use in nanomedicine and other areas of biotechnology.

## Figures and Tables

**Figure 1 biology-14-00950-f001:**
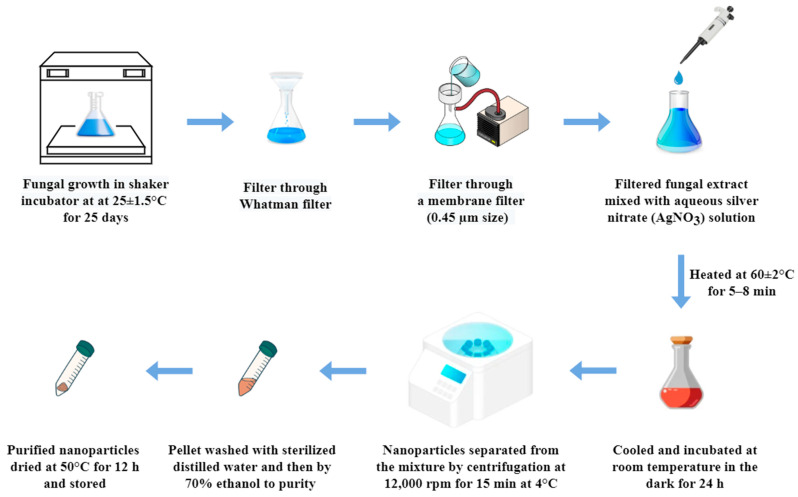
Schematic representation of fungal-assisted synthesis of silver nanoparticles (AgNPs).

**Figure 2 biology-14-00950-f002:**
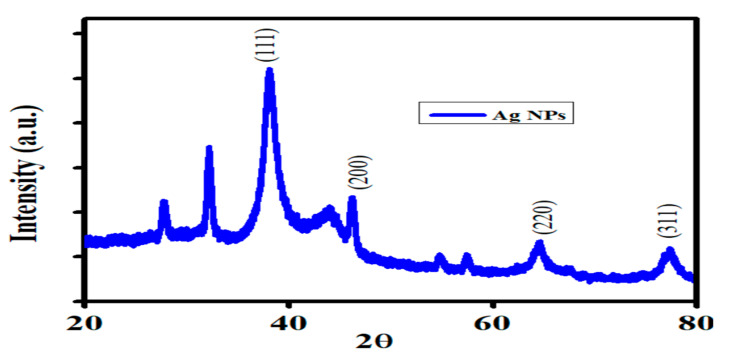
XRD of fungal-mediated synthesized AgNPs.

**Figure 3 biology-14-00950-f003:**
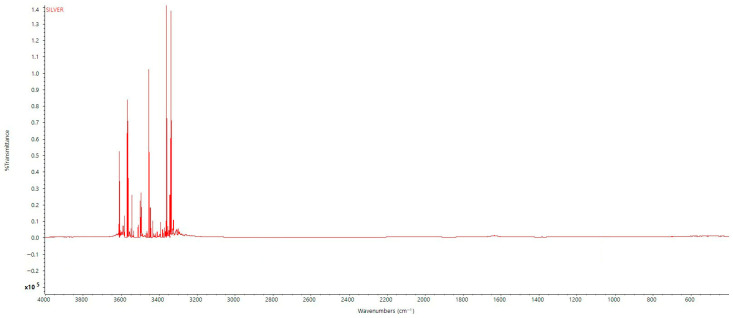
FTIR chromatogram of fungal-assisted synthesized silver nanoparticles.

**Figure 4 biology-14-00950-f004:**
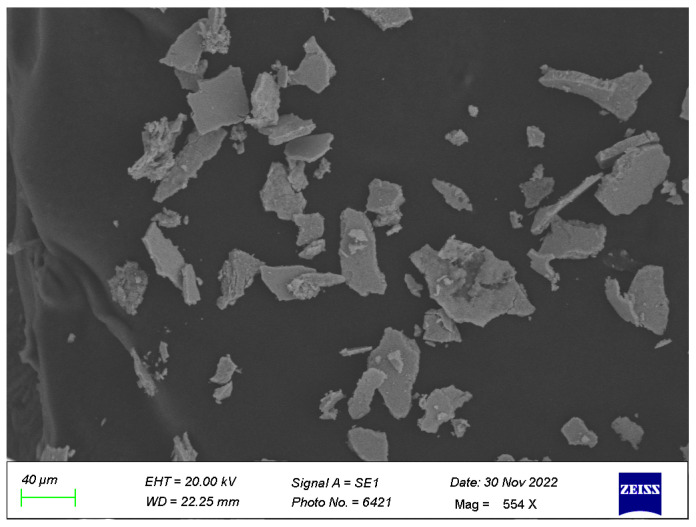
SEM micrograph of FANPs synthesized from 1 mM AgNO_3_ with fungal extract 40 µm magnification.

**Figure 5 biology-14-00950-f005:**
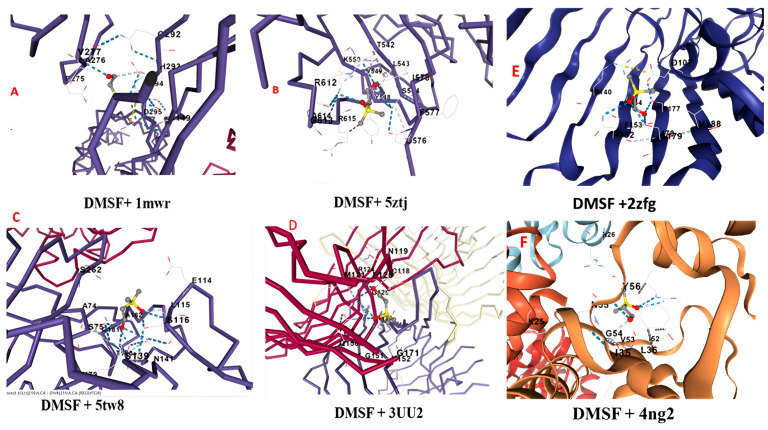
Molecular docking: (**A**). DMSF + 1mwr (*S. aureus*); (**B**). DMSF + 5ztj (*S. typhi*); (**C**). DMSF + 5tw8 (*S. aureus*); (**D**). DMSF + 3uu2 (*S. typhi*); (**E**). DMSF + 2zfg (*E.coli*); (**F**). DMSF + 4ng2 (*Pseudomonas*).

**Figure 6 biology-14-00950-f006:**
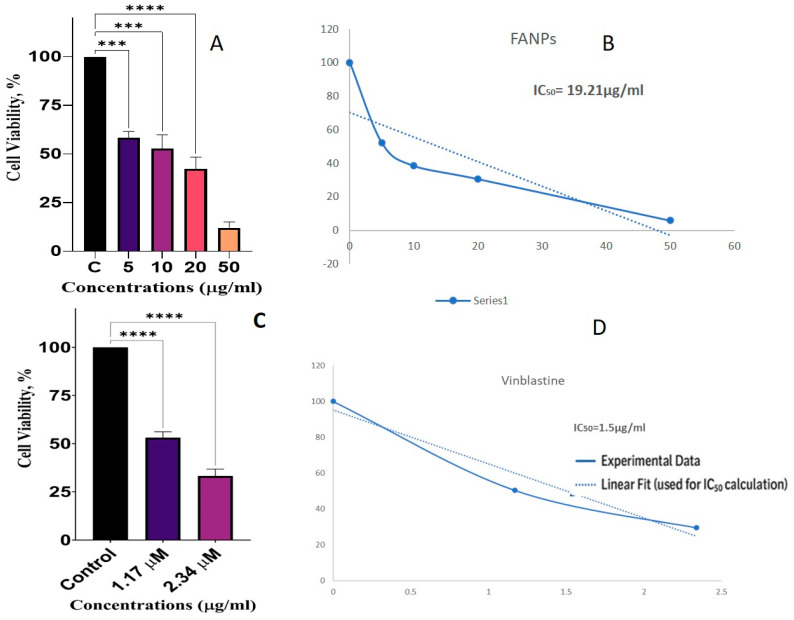
Anticancer activity of FANPs against the lung cancer cell line A549. (**A**) cell cytotoxicity with FANP, (**B**) cytotoxicity curve indicating IC_50_ 19.41μg/mL, (**C**) cell cytotoxicity with Vinblastin positive control (**D**) cytotoxicity curve indicating IC_50_ 1.2 μM, *** *p* < 0.005, **** *p* < 0.001.

**Figure 7 biology-14-00950-f007:**
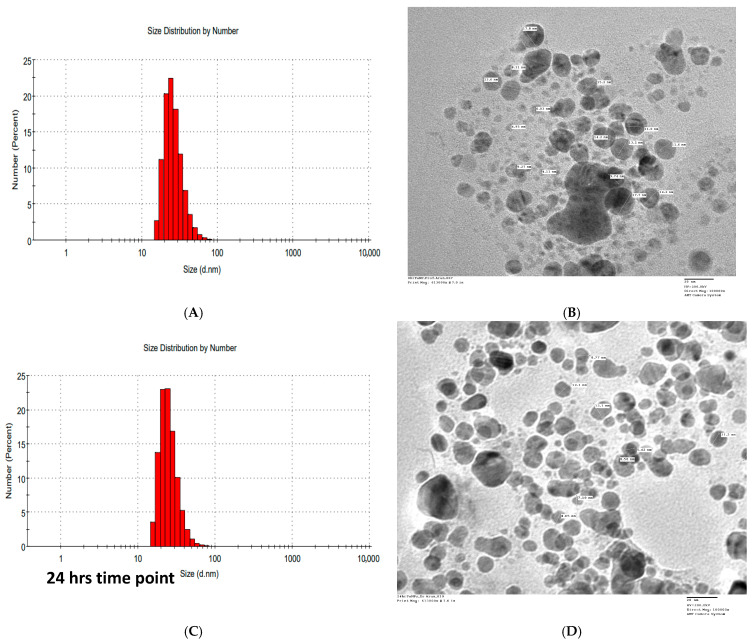
Dynamic light scattering demonstrates that the nanoparticles maintained their size and exhibited no aggregation after the different incubation period (0 h, 24 h, and 48 h DLS); (**A**) nanoparticle size distribution by DLS at 0 h, (**B**) nanoparticles TEM to determine shape and measuring size in nm at 0 h, (**C**) nanoparticle size distribution by DLS at 24 h, (**D**) nanoparticles TEM to determine shape and measuring size in nm at 24 h, (**E**) nanoparticle size distribution by DLS at 48 h, (**F**) nanoparticles TEM to determine shape and measuring size in nm at 48 h.

**Table 1 biology-14-00950-t001:** Comparison of Particle Size Results from Different Characterization Techniques.

S. No	Characterization Technique	Average Particle Size	Interpretation
1	XRD (Scherrer equation)	18.6 nm	Crystallite size only; based on (111) diffraction peak
2	TEM	11.0–20.4 nm	Physical core size; direct visualization of shape and distribution
3	DLS	~100 nm	Hydrodynamic diameter; includes capping agents and surrounding media

**Table 2 biology-14-00950-t002:** The list of phytoconstituents present in the FANPs using GC–MS.

Peak	Name	Area%	R.Time	Base *m*/*z*	Biological Activity	References
1	Dimethylsulfoxonium formylmethylide	98.88	3.566	63.15	Antimicrobial, antioxidant, and anticancer	[65,66]
2	Hexadecanoic acid	0.61	18.141	74.05	Antimicrobial, anti-proliferative effects	[67,68]
3	9-Octadecenoic acid	0.51	19.812	55.10	Antimicrobial, antioxidant, and anticancer	[69]

**Table 3 biology-14-00950-t003:** Antibacterial activity of the FANPs against bacterial strains.

S.No.	Name of Microorganism	Gram Reaction	Zone of Inhibition in mm			
			DMSO	Gentamicin	Chloramphenicol	AgNPs
1	*Escherichia coli*	GNB	NA	10.9	23.5	15.7 ± 0.07
2	*Salmonella typhi*	GNB	NA	21.2	23.0	18.1 ± 0.07
3	*Klebsiella pneumoniae*	GNB	NA	20.4	29.3	15.5 ± 0.28
4	*Pseudomonas aeruginosa*	GNB	NA	11.3	28.4	16.2 ± 0.14
5	*Bacillus cereus*	GPB	NA	19.8	26.6	15.0 ± 0.14
6	*Staphylococcus aureus*	GPB	NA	27.3	26.1	20.8 ± 1.20
7	*Bacillus subtilis*	GPB	NA	27.2	36.6	18.9 ± 0.35
8	*Streptococcus pneumoniae*	GPB	NA	23.2	24.3	17.6 ± 0.28

DMSO: Dimethyl sulfoxide; GPB: Gram-positive bacteria; GNB: Gram-negative bacteria; NA: No activity.

**Table 4 biology-14-00950-t004:** Molecular docking of DMSF with the Protein of *S. aureus* and *S. typhi*.

Ligand + Protein	Cur Pocket Id	Vina Score (Binding Affinity)	Cavity Score	Centre (x, y, z)	Docking Size (x, y, z)
DMSF + 5tw8	C1	−4.0	1541	33, −60, 5	22, 16, 33
DMSF + 1mwr	C1	−3.8	16,608	20, 34, 37	35, 34, 28
DMSF + 3uu2	C1	−3.9	987	−34. −35, 11	16,16,16
DMSF + 5ztj	C2	−3.5	123	23, 35, 35	16, 16, 16
DMSF + 2zfg	C3	−3.8	699	−12, 41, 4	16, 16, 16
DMSF + 4ng2	C3	−3.8	2457	5, 50, 8	24, 28, 35

DMSF: Dimethylsulfoxonium formylmethylide; 5tw8: Penicillin-binding protein 4 (PBP4) of *S. aureus*; 1mwr: Penicillin-binding protein 2a from methicillin resistant *S. aureus* strain 27r; 3uu2: OmpC is an outer membrane porin of *S. typhi*; 5ztj: Gyrase A C-terminal domain from *Salmonella typhi*; 2zfg: *E. coli* Outer membrane porin (OmpF); 4ng2: Pseudomonas LasR LBD-QslA complex.

## Data Availability

All data related to this research have been given in the manuscript and Appendix A.

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
