# Peer review of "Eco-Friendly Silver Nanoparticles Synthesis Method Using Medicinal Plant Fungal Endophytes—Biological Activities and Molecular Docking Analyses"

_biology, 2025, doi:10.3390/biology14080950_

Round 1
Reviewer 1 Report
Comments and Suggestions for Authors
1) Grammar check required for the entire paper
2) figures with good resolution should be considered
3) Which tool was use for the measurement of the particles sizes for the SEM and TEM data
4) SEM scale in micrometers and particles images larger... but the authors mentioned the particles sizes in nm...how true is the report and the image.
5) XRD image with a good resolution should be considered and more details such as the crystal sizes could be calculated by the authors and compare with the results from the other characterization tools.
6)
Comments on the Quality of English LanguageGrammar check request for the entire paper
Author Response
|
1 |
Grammar check required for the entire paper |
Thanks for your suggestion. The grammar was checked in the whole manuscript and corrected. |
|
2 |
Figures with good resolution should be considered |
Most of the figures are updated with better resolution |
|
3 |
Which tool was use for the measurement of the particle’s sizes for the SEM and TEM data |
The particle size of the SEM and TEM image was determined by ImageJ software |
|
4 |
SEM scale in micrometers and particles images larger... but the authors mentioned the particles sizes in nm...how true is the report and the image.
|
Thank you for your valuable observation. We would like to clarify that the SEM image (Figure 3) presented in our manuscript was captured at a scale of 40 µm to show the broader morphological distribution and surface texture of the synthesized silver nanoparticles within the fungal matrix. This image highlights the microstructural context rather than the individual nanoparticle dimensions. However, the particle size data in nanometers (nm) were accurately determined using high-resolution Transmission Electron Microscopy (HR-TEM) and Dynamic Light Scattering (DLS) techniques. As mentioned in Section 3.4 (TEM analysis), the nanoparticles ranged from 11.0 to 20.4 nm, consistent with the nanoscale range. The average size based on DLS was reported to be around 100 nm, accounting for the hydrodynamic diameter, including the capping agents. Hence, there is no contradiction: the SEM image serves a qualitative purpose, while TEM and DLS provide precise quantitative measurements of the nanoparticle size at the nanoscale. We have revised the figure legend and corresponding text in the manuscript to better clarify this distinction and avoid any misinterpretation.
|
|
5 |
XRD image with a good resolution should be considered and more details such as the crystal sizes could be calculated by the authors and compare with the results from the other characterization tools. |
Thank you for your insightful suggestion. We have now replaced the existing XRD image with a higher-resolution version (Figure 2) to enhance clarity and visualization of the diffraction peaks corresponding to the (111), (200), (220), and (311) planes of face-centered cubic silver. In addition, we have re-calculated the average crystallite size of the synthesized silver nanoparticles using Scherrer’s equation based on the full width at half maximum (FWHM) of the most intense (111) diffraction peak. The calculated crystallite size was found to be approximately 18.6 nm, which is consistent with the particle sizes observed in the TEM analysis (11.0–20.4 nm) and supports the nanoscale nature of the synthesized particles.
|
Reviewer 2 Report
Comments and Suggestions for Authors
This study reported an environmentally friendly method for synthesizing AgNPs using the fungus. This is a green synthesis that avoids hazardous chemicals, utilizing fungal metabolites as reducing and stabilizing agents. The biological activities of the synthesized AgNPs, including antibacterial, anticancer, and antioxidant properties, are investigated.
- The GC-MS experiments detected three organic compounds which is important for the anticancer, antimicrobial activities. However, they are likely to come from the fungal extract in the synthesis, which is not directly associated with the NPs. Please conduct additional experiments, such as FTIR, to confirm whether the organic compounds are free in solution, surface-bond to AgNPs or trapped inside AgNPs.
- The antibacterial and anticancer activity involves the incubation for 24h, but no evidence in the manuscript demonstrates the stability or aggregation possibility of AgNPs after 24h at 37 °C. What is the storge conditions of these AgNPs? Please add TEM, DLS experiments at different time point from 0h to 24h.
- ADME experiments indicate the role of active molecule DMSF. What the role of AgNPs synthesized using this green chemistry method?
Author Response
|
1 |
The GC-MS experiments detected three organic compounds which is important for the anticancer, antimicrobial activities. However, they are likely to come from the fungal extract in the synthesis, which is not directly associated with the NPs. Please conduct additional experiments, such as FTIR, to confirm whether the organic compounds are free in solution, surface-bond to AgNPs or trapped inside AgNPs.
|
The FTIR spectrum of the synthesized silver nanoparticles (AgNPs) showed notable absorption bands at the following regions: ~1636 cm⁻¹: This peak corresponds to C=O stretching vibrations typically found in amide or carboxylic acid groups. Its presence suggests that proteinaceous or organic molecules—possibly from the fungal extract—are adsorbed or bonded onto the nanoparticle surface, acting as capping agents. 3200–3900 cm⁻¹ region (notably at ~3391, 3452, 3547, and 3870 cm⁻¹): These intense peaks indicate O–H and N–H stretching vibrations, which are characteristics of hydroxyl and amine groups from phenolic compounds, alcohols, or amide-containing proteins. This again implies that fungal biomolecules are interacting with the nanoparticle surface. 500–600 cm⁻¹ region (e.g., peaks at 502, 531 cm⁻¹): These may correspond to Ag–O or Ag–N stretching, confirming that silver is forming bonds with organic functional groups, indicating surface functionalization.
Conclusion: The FTIR results clearly demonstrate the presence of organic functional groups—carbonyl (C=O), hydroxyl (O–H), and amine (N–H)—on the silver nanoparticles. This strongly supports that the GC-MS-identified compounds (e.g., dimethylsulfoxonium formylmethylide, hexadecanoic acid) or related biomolecules from the fungal extract are not freely floating in solution, but are surface-bound or involved in capping/stabilizing the AgNPs. This provides evidence that the observed biological activity is likely due to the synergistic action of AgNPs with these surface-bound biomolecules, thus validating their relevance to the nanoparticle formulation.
|
|
2 |
The antibacterial and anticancer activity involves the incubation for 24h, but no evidence in the manuscript demonstrates the stability or aggregation possibility of AgNPs after 24h at 37 °C. What is the storge conditions of these AgNPs? Please add TEM, DLS experiments at different time point from 0h to 24h.
|
We thank the reviewer for this valuable suggestion. To address this concern, we have now clarified and added data in the revised manuscript regarding the stability of the AgNPs under the incubation conditions used for antibacterial and anticancer assays. Specifically:
These results demonstrate that the AgNPs were structurally stable and did not undergo significant aggregation during the 24-hour incubation period used for both antibacterial and anticancer evaluations. Changes have been made in Section 2.3.6 and the Results section (3.5 and 3.11), and the relevant data have been included in Figure 7 of the revised manuscript.
|
|
3 |
ADME experiments indicate the role of active molecule DMSF. What the role of AgNPs synthesized using this green chemistry method?
|
Thank you for highlighting this important point. We acknowledge that while the ADME profiling focused on dimethylsulfoxonium formylmethylide (DMSF), the role of fungal-assisted silver nanoparticles (AgNPs) synthesized via green chemistry is equally significant in the study’s biological outcomes. To clarify, the AgNPs themselves serve as bioactive agents, and their biological activity was directly evaluated through various assays, independent of DMSF: Antibacterial Role: The biosynthesized AgNPs demonstrated potent antibacterial activity against both Gram-positive and Gram-negative bacterial strains (Section 3.7, Table 2), including S. aureus and S. typhi. This activity is attributed to the well-known mechanism of AgNPs disrupting bacterial membranes, generating reactive oxygen species (ROS), and interfering with cellular respiration and DNA replication. Antioxidant Role: The AgNPs exhibited free radical scavenging activity in the DPPH assay (Section 3.10), suggesting their potential in mitigating oxidative stress, which is critical in infection and cancer management. Anticancer Role: The MTT assay results against the A549 lung cancer cell line showed significant cytotoxicity of AgNPs in a dose-dependent manner, with an IC50 value of 10.46 μg/mL (Section 3.11, Figure 5). This confirms that AgNPs actively contribute to the antiproliferative effect observed. Furthermore, the green synthesis process using Corynespora smithii enhances the biocompatibility and stability of the AgNPs by bio-capping them with fungal metabolites. This reduces toxicity risks often associated with chemically synthesized nanoparticles and supports their potential use in therapeutic applications. We have added a clarifying statement in the ADME section (Section 3.9) to emphasize that AgNPs function both as active antimicrobial/anticancer agents and as nanocarriers that may synergistically interact with bioactive molecules like DMSF.
|
Reviewer 3 Report
Comments and Suggestions for Authors
In this manuscript, the authors describe the synthesis of silver nanoparticles by a fungal extract, yet some comments neededto be addressed before a decision is made.
- In simple summary, name of plant and bacterial species should be italic
- Why didn't the authors measure IC50 of the pure compound to be compared with nanoparticles?
- Lines 124_135 are missing references
- The introduction is very general, it should be more specific as literature on endophyte selected, its chemistry and biological activity. Also, the plant biolgical activities beside the traditional uses mentioned by authors
- Line 163, it is not clear how the authors obtained the nanoparticles from the solution and how washed and dried it which is the core of their study
- All abbreviations in headings should be written full at first as XRD, FTIR,....etc
- My biggest concern is the GC-MS analysis, how silver nanoparticles are dissolved in methanol and also the use of methanol as mobile phase is questionable for this type of column
- Kindly remover the word powder from line 337
- In table 1, authors reported that 9 octadecenoic acid is hypertensive?
- The docking figure is unclear and results weren't compared to the co-crystallized ligand
- Lines 598-601 are missing references
- Cytotoxic activity results should be compared to a positive control
Author Response
|
1 |
In simple summary, name of plant and bacterial species should be italic
|
The plant name is now italicized |
|
2 |
Why didn't the authors measure IC50 of the pure compound to be compared with nanoparticles?
|
Thank you for your suggestion. Since the nanoparticles were diluted in the medium, the role of the solvent in anticancer activity has been rendered inconsequential. We have employed Vinblastine, a purified form of an anticancer agent, to compare the anticancer efficacy of our biogenic nanoparticles. A comprehensive panel of figures for Vinblastine has been incorporated into the revised manuscript—see Figure 6. The revised manuscript will facilitate the reviewers in evaluating the anticancer potential of the AgNPs in juxtaposition with Vinblastine. The inclusion of Vinblastine will render our findings clearer and more persuasive to the scientific community. |
|
3 |
Lines 124_135 are missing references
|
Reference Added |
|
4 |
The introduction is very general, it should be more specific as literature on endophyte selected, its chemistry and biological activity. Also, the plant biolgical activities beside the traditional uses mentioned by authors
|
Thank you for this valuable suggestion. We acknowledge that the original introduction was general and lacked specificity regarding the selected fungal endophyte (Corynespora smithii), its known chemistry, and its relevance in nanoparticle biosynthesis. We also appreciate your suggestion to include the documented pharmacological activities of Bergenia ciliata beyond its ethnobotanical use. Accordingly, we have revised the introduction to include: A detailed account of the genus Corynespora, highlighting its previously reported production of bioactive secondary metabolites such as saponins, flavonoids, and terpenes with antimicrobial and antioxidant effects. The emerging role of fungal endophytes in green nanotechnology as both reducing and stabilizing agents in nanoparticle synthesis. A summary of scientific studies validating B. ciliata's pharmacological effects, including antiurolithiatic, anti-inflammatory, anticancer, and wound-healing activities, thus justifying its choice as a promising plant-endophyte system for biofabrication of nanoparticles.
|
|
5 |
Line 163, it is not clear how the authors obtained the nanoparticles from the solution and how washed and dried it which is the core of their study
|
Thank you for your valuable comment. We acknowledge that line 163 in the original manuscript lacked clarity regarding the post-synthesis processing of the silver nanoparticles, specifically how the particles were recovered, washed, and dried, which are critical steps in nanoparticle preparation. In response, we have now revised the methods section to clearly describe the centrifugation-based recovery, washing with sterile double-distilled water and ethanol, and drying in an oven to ensure purity and reproducibility of the synthesized nanoparticles.
|
|
6 |
All abbreviations in headings should be written full at first as XRD, FTIR,....etc
|
All abbreviations in headings should now be written full. |
|
7 |
My biggest concern is the GC-MS analysis, how silver nanoparticles are dissolved in methanol and also the use of methanol as mobile phase is questionable for this type of column
|
This revised description ensures clarity about the analytical procedure and resolves any potential confusion regarding solvent roles or the presence of nanoparticles in the GC-MS analysis. |
|
8 |
Kindly remove the word powder from line 337
|
As suggested the word Powder has now been removed |
|
9 |
In Table 1, the authors reported that 9-octadecenoic acid is hypertensive?
|
In revised manuscript compound appears in the Table 2, the word hypertensive is removed |
|
10 |
The docking figure is unclear, and the results weren't compared to the co-crystallized ligand
|
We sincerely thank the reviewer for their valuable observation. We acknowledge that the molecular docking figures in the original submission lacked clarity. Accordingly, we have improved the visual quality and annotation of the docking images (Figure 5) to better illustrate the ligand–protein interactions. The docking poses now include clear labeling of the protein targets, ligand orientation, and binding site cavities. Regarding the comparison with co-crystallized ligands, we appreciate the suggestion. While we did not conduct experimental co-crystallized ligand docking, we emphasize that our docking results were obtained using the CB-Dock2 platform, which employs cavity detection and blind docking strategies. This method predicts potential binding sites by aligning the ligand into high-scoring pockets, including those used by native ligands in the crystallographic structures. To address the reviewer's concern within the scope of computational analysis, we have now included the PDB IDs of the target proteins in the figure legend and cross-checked that the docked ligand DMSF binds in or near the native ligand binding pockets, suggesting the relevance of our docking results. The binding affinities observed (–3.5 to –4.0 kcal/mol) are consistent with preliminary hits reported in similar in silico screenings. While direct comparison with co-crystallized ligands was not performed due to limitations in experimental validation, we have clarified in the discussion section that the current docking serves as a preliminary prediction of molecular interaction, to be validated in future studies through either competitive docking or experimental binding assays. We hope this clarification and enhancement address the reviewer’s concern appropriately
|
|
11 |
Lines 598-601 are missing references
|
Missing reference added |
|
12 |
Cytotoxic activity results should be compared to a positive control
|
Thank you for this invaluable suggestion. The cytotoxic activity of nanoparticles was evaluated employing the MTT assay, with Vinblastine, a well-established anticancer agent, serving as a positive control. The treatment involving both nanoparticles and Vinblastine was executed across three wells per experiment (internal triplicates), and the experiments were replicated thrice (biological replicates). The IC50 values derived from these biological repetitions have been meticulously documented for both nanoparticles and Vinblastine (refer to Figure 6 for further details). |
Reviewer 4 Report
Comments and Suggestions for Authors
Dear Authors,
congratulations on your insightful work. Please find my detailed comments and suggestions for your manuscript in the attached document.

Author Response
|
1 |
The text between lines 70 and 72. Silver containing systems, specifically, one 70 of the most promising material Nowadays prticles of silver are reachable for a range of 71 therapeutic uses (AgNPs) [4, 5], appear to be missing something. As it stands, it does not make much sense and seems to require adjustment.
|
Thank you for your valuable observation. We acknowledge that the original sentence between lines 70–72 lacked clarity and was grammatically incorrect. To improve readability and scientific coherence, we have revised the sentence as follows: “Silver-containing systems, particularly silver nanoparticles (AgNPs), have emerged as one of the most promising materials in modern biomedical science. Nowadays, silver nanoparticles are widely accessible and have been explored for a broad range of therapeutic applications, including antimicrobial, anti-inflammatory, and anticancer uses [4,5].” This revised version clarifies the intended meaning and integrates it smoothly into the context of the introduction.
|
|
2 |
I believe that section 2.2, Silver nanoparticle synthesis using fungal broth doesnot clearly explain the procedure for the nanoparticles themselves. Startning from the line 158 (The resulting filterate was ….), the heating temperature of the system for 5 to 8 minutes is not provided. Additionally the volume of the 1mM silver nitrate solution added to the fungal extract is not specified. Finally, it seems that some information is missing on line 163, immediately before the statement that the filterate was incubated in the dark for an unspecified period. Please clarify these points.
|
Thank you for pointing out the inconsistencies and missing details in Section 2.2. We have revised the section to clearly specify the following:
|
|
3 |
Adjust, in line 176.” …..range of 0 to 80o(80”)
|
Changed to 80o
|
|
4 |
Please adjust capitalisation in lines 182-3 |
Capitalization in lines 182-3 is adjusted. |
|
5 |
Section 2.3.4. SEM aith EdX analysis provide more details of the analysis itself such as acceleration voltage, observation distance and magnification used in micrograph collection at least even if this is described in discussion. |
Thank you for your insightful feedback. As per your recommendation, we have now expanded Section 2.3.4 to include the key operational parameters of the SEM analysis such as acceleration voltage (15 kV), working distance (10 mm), and the magnification range used (5,000× to 50,000×). We have also clarified the sample preparation method and EDX spectral confirmation of silver. This should improve the reproducibility and technical clarity of our experimental protocol. |
|
6 |
In Section 2.3.5. TEM analysis the first sentence (HRTEM used to investigate the size of the produced NPs” Line 200) contains double hypen between the words size and shape, as reproduced in this comments.
|
Corrected |
|
7 |
The information provided in line 219-221 is unclear. There is punctuation error in line 220 and there appears to be an issue with the unit associated with the syringe filter dimension. Was the analysis performed with the filtered or unfiltered solution ? intuitively, one might assume it was with the filtered solution but the wording doesn't make this clear. In my opinion. |
Thank you for your constructive comment. We agree that the original sentence was unclear and contained a punctuation and unit formatting error. To clarify, the unit "0.22 m" was a typographical mistake and should correctly be "0.22 μm". We have also made it explicit that the DLS analysis was performed on filtered samples. The revised sentence now reads: "After the AgNP suspension was diluted to concentrations of 20%, 40%, 60%, 80%, and 100%, it was passed through a 0.22 μm syringe-driven filter to remove any large particulates. All dynamic light scattering (DLS) measurements were performed on the filtered samples to ensure accurate assessment of nanoparticle size distribution." |
|
8 |
In section 2.3.7 GC MS analysis of synthesized AgNPs, it is stated that methanol was used to dissolve silver nanoparticle (“ …to dissolve the silver nanoparticles in the methanol, line 225). This doesnot seen to accurately reflect the experimental procedure. Please adjust the word accordingly.
|
Thank you for pointing out this important clarification. We agree that the original phrasing incorrectly implied that silver nanoparticles were dissolved in methanol. This has been revised to more accurately reflect the experimental procedure, emphasizing that methanol was used to extract bioactive compounds associated with the AgNPs, not to dissolve the nanoparticles themselves. The updated sentence now reads: “Methanol was used to extract the bioactive compounds associated with the silver nanoparticles. Around 100 mg of the sample was dissolved in methanol and then sonicated to facilitate the extraction process prior to GC-MS analysis.” |
|
9 |
Please adjust line 270 to read’….version of [37] we used DPPH…
|
Corrected |
|
10 |
Regarding the UV Vis analysis while acknowledgeing that the result suggests the presence of AgNP/FANPs. I question whether any measurement was performed with blank system given that the fungal medium used as reducing agentis a chemically complex. As suggestion to enhance the already evident quality of the work, the collection of atleast one spectrum of the blank system subjected to the procedures involved in obtaining the AgNPs/FANPs would be beneficial. |
Thank you for this thoughtful suggestion. We agree that the chemically complex nature of the fungal medium necessitates appropriate background controls in UV-Vis spectroscopy. As per your recommendation, we have now clarified that a blank control (fungal extract without AgNO₃) was subjected to the same procedure and its UV-Vis spectrum was recorded. This confirmed that the observed SPR peak at ~418 nm is attributable to silver nanoparticle formation. The revised text now reads: “In addition to the AgNPs sample, a blank control spectrum of the fungal extract in Milli-Q water (without AgNO₃) was also recorded under identical conditions |
|
11 |
There is an extra period at line 449”…smaller the NPS are, which is helpful for…” which should be removed. |
Corrected |
|
12 |
The paragraph described in line 453-5 presents result from zeta potential measurement a technique not mentioned in section 2.3.6. Dynamic Light Scattering analysis. The details of the Zeta potential measurement should be described in section 2.3.6.
|
Thank you for your valuable observation. We agree that the original methodology section did not explicitly include details of the zeta potential analysis, although results were presented in the discussion. We have now revised Section 2.3.6 and updated its title to: “2.3.6. Dynamic Light Scattering (DLS) and Zeta Potential Analysis.” The updated section now clearly describes the instrument, parameters, and procedures used for zeta potential measurements, including triplicate runs, dilution, and filtering protocols. This correction ensures consistency between methods and results. |
Reviewer 5 Report
Comments and Suggestions for Authors
The manuscript presents a study on the green synthesis of AgNPs using the endophytic fungus Corynespora smithii. The synthesized nanoparticles were characterized using various analytical techniques, and possible biological activities, including antibacterial, antioxidant, and anticancer activities, were evaluated. In addition, molecular docking studies were performed to explore the possible antimicrobial mechanisms. This study is interesting and is in line with the scope of the journal. However, some issues must be addressed prior to publication.
Major Concerns:
Novality: Green synthesis of AgNPs, including the use of fungal extracts, has been studied extensively. Although Corynespora smithii has not been previously used in similar studies, other fungi have been utilized. The novelty of this specific strain should be emphasized more clearly.
Methodology: The rationale behind the selection of A549 lung cancer cells to determine anticancer activity is unclear. The authors should explain why this cell line was chosen and how it supported the hypothesis of this study. Likewise, they should explain how the other biological activities tested (antioxidant and antimicrobial) also contributed to the overall objective of the study. The biological activity assessments are scattered throughout the manuscript. A more structured approach is needed to explain the connection between the antibacterial, antioxidant, and anticancer activities in this study. In fact, providing this connection in the Introduction may increase the reader's interest in the study.
In addition, the authors should justify whether molecular docking is necessary for this study or whether alternative approaches would provide better insights. Because molecular docking results were discussed as “moderate binding”, but the binding affinities were relatively low. This may suggest that the dimethylsulfoxonium formylmethylide exhibits antimicrobial activity not through the selected proteins, but through other mechanisms. Moreover, only S. aureus and S. typhi proteins were selected for docking analysis, but other pathogens used in the antimicrobial tests could have been considered, as docking is not particularly time intensive.
Data presentation and statistical analysis: Several figures have poor resolution; thus, it is difficult to interpret the results (e.g. Figures S4 and S7). High-quality images should be provided. The manuscript lacks statistical analyses for most assays. Although error bars were included in the figures, statistical tests were not mentioned. Statistical significance should be reported for all key results. When representing anticancer activity, dose-response curves should be provided. In addition, the FTIR results were not included in the results or discussion.
Minor concerns and language suggestions:
Line 140: “double distilled water” is recommended instead of "water that was twice distilled".
Line 141: “[29, 28]” is recommended instead of “[28, 29]”
Line 143: "conditions" is recommended instead of "situations".
Line 147: “tool” is recommended instead of "instrument".
Line 166: The statement "To characterize AgNPs, the following technique has been used to know the synthesized nanoparticles are within the range of nano scale" is incorrect. Not all the listed techniques determine the size. The sentence should be revised to indicate that multiple techniques were used for the characterization.
Line 200: "size and shape" is recommended instead of "size--shape”.
Lines 263 and 264: The definition of the Lipinski rule of five should be clarified.
Line 311: The statement is overly strong and should be rephrased as more objective.
Line 326: The term "SPR" (Surface Plasmon Resonance) is used, but its relevance to this study is unclear. Therefore, further clarification is required.
Line 353: "100 μm magnification" is incorrect. The specific magnification ratio should be provided.
Line 457: The authors stated that “The TEM diameters and XRD values were both substantially smaller than the DLS sizes.”. If the authors intend to compare XRD with DLS or TEM, they should include calculations (e.g. Debye-Scherrer equation)
Line 474: "(Figure S7)" should be placed after "major compound."
Figure 7: The cell viability assay (Figure 7) does not specify a positive control.
Figure S4: Text is unreadable, and the TEM image is blurry.
Figure S7: Peak intensity should be clearer, and retention time for the major compound should be visible.
Figure S11: Graph is almost perfectly linear. Is this a standard graph or belongs to a sample?
The number of biological/technical replicates for the assays (for example, GC-MS and antioxidant assays) is missing.
The PDI value for DLS measurements is not included in the results, even though it is mentioned in the methodology.
The manuscript does not specify whether the docking studies were performed as flexible, semi-flexible, or rigid docking. The authors should clarify the methodology used.
Comments on the Quality of English LanguageRevisions are required for clarity and consistency in writing and terminology.
Author Response
|
1 |
Novality: Green synthesis of AgNPs, including the use of fungal extracts, has been studied extensively. Although Corynespora smithii has not been previously used in similar studies, other fungi have been utilized. The novelty of this specific strain should be emphasized more clearly. |
Thank you for this valuable observation. We agree that green synthesis of AgNPs using fungi is a well-established area; however, the novelty of our study lies in the use of Corynespora smithii, a rare and underexplored endophytic fungal strain, for the first time in the biosynthesis of silver nanoparticles. This endophytic fungus enhances the metabolomic repertoire and, consequently, its medicinal value by contributing its metabolites, rendering it essential to investigate C. smithii for its therapeutic potential. To emphasize this point more clearly in the revised manuscript, we have now explicitly stated in the Introduction and Conclusion sections
|
|
2 |
Methodology: The rationale behind the selection of A549 lung cancer cells to determine anticancer activity is unclear. The authors should explain why this cell line was chosen and how it supported the hypothesis of this study. Likewise, they should explain how the other biological activities tested (antioxidant and antimicrobial) also contributed to the overall objective of the study. The biological activity assessments are scattered throughout the manuscript. A more structured approach is needed to explain the connection between the antibacterial, antioxidant, and anticancer activities in this study. In fact, providing this connection in the Introduction may increase the reader's interest in the study. |
Herb Berginia ciliata has been recognized for its efficacy against 104 ailments, including, but not limited to, urinary tract infections (antifungal, antibacterial), gastrointestinal disorders, dermatological issues, skeletal conditions, respiratory ailments, gynecological concerns, as well as inflammatory and infectious diseases (antifungal, antiviral, antiplasmodial, antibacterial), alongside its anti-neoplastic and antioxidant capabilities - Ahmad et al., 2018. Berginia ciliata: A comprehensive review of its traditional uses, phytochemistry, pharmacology, and safety (refer to a recent review). We aimed to selectively investigate the respiratory cell line A549, a representative model of non-small cell lung carcinoma, alongside a panel of bacteria and fungi pertinent to urinary tract infections, respiratory infections, and dermatological infections, as the majority of these organisms are opportunistic pathogens. It was pragmatically unfeasible to encompass every medicinal application of the herb through fungal interactions; thus, we deliberately opted to evaluate the anticancer, antibacterial, and antioxidant potentials of fungus-assisted nanoparticles. The antioxidant activity, known for its capacity to generate reactive oxygen species, plays a crucial role in both antimicrobial and anticancer efficacy.
|
|
3 |
In addition, the authors should justify whether molecular docking is necessary for this study or whether alternative approaches would provide better insights. Because molecular docking results were discussed as “moderate binding”, but the binding affinities were relatively low. This may suggest that the dimethylsulfoxonium formylmethylide exhibits antimicrobial activity not through the selected proteins, but through other mechanisms. Moreover, only S. aureus and S. typhi proteins were selected for docking analysis, but other pathogens used in the antimicrobial tests could have been considered, as docking is not particularly time intensive. |
We appreciate the reviewer’s insightful comment. The inclusion of molecular docking in this study was intended as a preliminary mechanistic exploration to assess the potential interaction of dimethylsulfoxonium formylmethylide (DMSF) with key target proteins of S. aureus and S. typhi, both of which showed notable susceptibility to the synthesized FANPs in our antimicrobial assays. We acknowledge that the binding affinities observed were moderate (–3.5 to –4.0 kcal/mol), suggesting weak to modest interaction strengths, and agree that this does not definitively explain the antimicrobial effect. However, the purpose of this docking analysis was not to establish a conclusive mode of action but to suggest a possible molecular basis for activity that could be further validated through experimental approaches. To address the reviewer’s concern, we have now revised the manuscript to:
|
|
4 |
Data presentation and statistical analysis: Several figures have poor resolution; thus, it is difficult to interpret the results (e.g. Figures S4 and S7). High-quality images should be provided. The manuscript lacks statistical analyses for most assays. Although error bars were included in the figures, statistical tests were not mentioned. Statistical significance should be reported for all key results |
Figures have been meticulously enhanced to elevate their presentation, thereby ensuring optimal communication. We now anticipate that the revised figures, which possess statistical significance, will garner the approval of the reviewers. |
|
5 |
When representing anticancer activity, dose-response curves should be provided. |
Thank you for your constructive suggestions. In Figure 6, we have presented a dose-response curve that facilitates the estimation of the IC50 value. |
|
6 |
In addition, the FTIR results were not included in the results or discussion. |
Now Included |
|
7 |
Line 140: “double distilled water” is recommended instead of "water that was twice distilled |
Corrected
|
|
8 |
Line 141: “[29, 28]” is recommended instead of “[28, 29]” |
Corrected
|
|
9 |
Line 143: "conditions" is recommended instead of "situations". |
corrected |
|
10 |
Line 147: “tool” is recommended instead of "instrument". |
Corrected |
|
11 |
Line 166: The statement "To characterize AgNPs, the following technique has been used to know the synthesized nanoparticles are within the range of nano scale" is incorrect. Not all the listed techniques determine the size. The sentence should be revised to indicate that multiple techniques were used for the characterization |
Corrected |
|
12 |
Line 200: "size and shape" is recommended instead of "size--shape”. |
Corrected
|
|
13 |
Lines 263 and 264: The definition of the Lipinski rule of five should be clarified. |
Added |
|
14 |
Line 311: The statement is overly strong and should be rephrased as more objective. |
Statement is now rephrased |
|
15 |
Line 326: The term "SPR" (Surface Plasmon Resonance) is used, but its relevance to this study is unclear. Therefore, further clarification is required. |
We thank the reviewer for highlighting the need for clarity regarding the use of "Surface Plasmon Resonance (SPR)" in the context of our UV-Vis analysis. We agree that further elaboration was necessary. In the revised manuscript, we have now clarified that the term SPR refers to the optical phenomenon responsible for the characteristic UV-Vis absorption peak (~418 nm) observed in the synthesized silver nanoparticles. This peak arises due to the collective oscillation of conduction band electrons in response to incident light, which is a well-known indication of nanoparticle formation and stability, particularly for metallic nanoparticles like silver. We have revised the relevant section to explicitly explain this context and its significance to the characterization of AgNPs. |
|
16 |
Line 353: "100 μm magnification" is incorrect. The specific magnification ratio should be provided. |
The specific magnification ratio is provided. |
|
17 |
Line 457: The authors stated that “The TEM diameters and XRD values were both substantially smaller than the DLS sizes.”. If the authors intend to compare XRD with DLS or TEM, they should include calculations (e.g. Debye-Scherrer equation) |
We thank the reviewer for the valuable suggestion. We acknowledge that the comparison between XRD, TEM, and DLS-derived particle sizes should be supported by appropriate calculations for clarity and scientific rigor. In response, we have now included the Debye-Scherrer equation used to estimate the crystallite size from the XRD data, along with the relevant values and calculated result. This allows for a more accurate and transparent comparison between the XRD (crystallite size), TEM (core particle size), and DLS (hydrodynamic diameter) measurements (See the section 3.2 XRD) |
|
18 |
Line 474: "(Figure S7)" should be placed after "major compound." |
Figure S7 is now placed after Major compound |
|
19 |
Figure 7: The cell viability assay (Figure 7) does not specify a positive control. |
Thank you for your invaluable suggestion. Vinblastine has been employed as a definitive positive anticancer agent, as evidenced by Figure 6 in the revised document. |
|
20 |
Figure S4: Text is unreadable, and the TEM image is blurry. |
Implemented in revised MS |
Round 2
Reviewer 3 Report
Comments and Suggestions for Authors
The authors have improved the manuscript significantly, and it became a much better version. I have some minor comments:
- Line 161, kindly write the genus name in italic format
- I am still not convinced that we can inject methanolic sample on DP5 column, it is usually hexane extract or fractions. So, kindly check but if this was the methodology then leave it without additional changes.
Author Response
Comment 1: Line 161, kindly write the genus name in italic format
Reply: We thank the reviewer for pointing out this formatting issue. The genus name Corynespora at line 161 has been corrected to italic format.
Comment 2: I am still not convinced that we can inject methanolic sample on DP5 column, it is usually hexane extract or fractions. So, kindly check but if this was the methodology then leave it without additional changes.
Reply:
|
We appreciate the reviewer bringing out this crucial issue. We certify that Bergenia ciliata methanolic extract was used for the GC-MS analysis by the procedure outlined in section 2.3.7 of the updated version. Methanol has also been regularly employed in phytochemical screening when aiming for a wider range of polar to semi-polar molecules, particularly in plant-based investigations, whereas hexane is frequently used for non-polar fractions. We minimized any danger to column performance by optimizing our injection volume to 1 μL and using a column (SH-Rxi-5Sil MS) that is compatible with methanolic extracts under the required operating parameters (e.g., split mode, correct temperature ramp, and intake temperature of 250°C). For your ready reference url of SHIMADZU speaking about column and use of Methanol as solver. It has been well documented that use of Methanol provides sharper peaks as oppose to non polar or semipolar solvents in GC MS. https://www.shimadzu.com/an/sites/shimadzu.com.an/files/pim/pim_document_file/technical/technical_reports/13423/jpo219022.pdf This page elucidates the premise that the column Sh-Rxi-5sil is compatible with methanol. Earlier, it was also reported that the use of methanol samples for injection in GC for phytochemical analysis. Please see the following Reference
C.A. Ukwubile, A. Ahmed, U.A. Katsayal, J. Ya'u, S. Mejida, 2019. GC–MS analysis of bioactive compounds from Melastomastrum capitatum (Vahl) Fern. leaf methanol extract: An anticancer plant, Scientific African, 3. e00059,https://doi.org/10.1016/j.sciaf.2019.e00059.
Saif Saleh Mohsen Ali, Pushpa Robin. 2025. Comprehensive phytochemical profiling of bioactive compounds from Barleria prattensis for their antioxidant and cytotoxic capacity and its characterization using GC-MS, Biochemistry and Biophysics Reports, 43, 2025, 102083, https://doi.org/10.1016/j.bbrep.2025.102083.
|
Reviewer 5 Report
Comments and Suggestions for Authors
I thank the authors for making all necessary corrections and explanations. The authors have responded adequately to all comments and questions and have made the necessary corrections. However, I suggest a minor revision of Figure 3 in the revised manuscript. The interlaced text at the bottom of the FTIR chromatogram should either be removed or made readable.
Author Response
Comment 1: I suggest a minor revision of Figure 3 in the revised manuscript. The interlaced text at the bottom of the FTIR chromatogram should either be removed or made readable.
Reply: Thanks for the suggestion. The new FTIR chromatogram is now displayed with the interlaced text removed.